# Localized carbon deposition enables trimming of photonic integrated circuits

Rongyang Xu [1,5], Zhongyu Tang[1,5], Liam McRae[1], Akhil Varri [2], Frank Brückerhoff-Plückelmann [1], Xinyu Ma[1], Julian Rasmus Bankwitz[1], Julius Römer[1], Ravi Pradip[1], Qinlin Zhang[1], Lennart Meyer [1], Zhe Zhao [2], Jelle Dijkstra [1], Harish Bhaskaran [3], Rasmus R. Schröder [4], Wolfram H. P. Pernice [1] ✉ & Shabnam Taheriniya[1] ✉

Photonic integrated circuits (PICs), widely used in optical communications and computing, require precise post-fabrication trimming due to their high sensitivity to fabrication imperfections. Focused ion beam (FIB) carbon deposition offers a localized trimming approach with high spatial precision. Here, we demonstrate this technique for the first time to enable non-volatile post-fabrication trimming of PICs. To validate this approach, we use asymmetric directional couplers as representative fabrication-sensitive components. Structural characterizations confirm localized surface deposition without observable modification of the underlying waveguide core, and device measurements show discrete transmission tuning levels of 1.46–16.1 dB. Independent test structures further reveal an additional loss of 0.35 dB per π phase shift, indicating a low optical penalty at the device level. Furthermore, the optical response remains stable over two months following a brief initial settling phase. These results highlight the potential of FIB carbon deposition in device-level trimming and provide a foundation for exploring future trimming strategies toward parallel implementations.

A photonic integrated circuit (PIC) is a microchip that brings together multiple photonic components with different functions. Due to the inherent properties of light, such as high speed, low propagation loss, and high parallelism, PICs have been widely explored in applications such as optical communications[1–4], quantum computing[5–7], and neuromorphic computing[8–10]. The performance of PICs in these applications relies on the proper operation of precisely engineered photonic structures, which are often vulnerable to fabrication imperfections. The accumulation of optical response deviations with increasing device count has thus made fabrication imperfections a key obstacle to the further development of PICs. Post-fabrication trimming provides an effective approach to correct such imperfections, offering precise, localized, and flexible control over photonic device performance[11–13].

Various post-fabrication trimming techniques have been proposed to address fabrication imperfections. Volatile approaches are typically based on thermal tuning using micro-heaters[14,15], with the trimmed state sustained while electrical power is applied. In contrast, non-volatile approaches, such as electron beam exposure of polymer claddings[16–18], programmable phase-change materials (PCMs) integrated on waveguides[19–22], laser-based trimming[23,24], high-temperature local annealing[25–27], and focused ion beam (FIB) processing[15,28–31], permanently modify the optical response without sustained power consumption, but rely on distinct physical mechanisms and are implemented under different processing conditions. Polymer cladding trimming is a simple and cost-effective method, although its long-term stability can be limited depending on environmental conditions. PCM-based methods are broadly compatible with multiple material

[1]Kirchhoff-Institute for Physics, Heidelberg University, Heidelberg, Germany. [2]Institute of Physics, University of Münster, Münster, Germany. [3]Department of Materials, University of Oxford, Oxford, UK. [4]BioQuant, Heidelberg University, Heidelberg, Germany. [5]These authors contributed equally: Rongyang Xu, Zhongyu Tang. ✉e-mail: wolfram.pernice@kip.uni-heidelberg.de; shabnam.taheriniya@kip.uni-heidelberg.de

platforms, but they generally introduce higher insertion loss due to mode mismatch and intrinsic absorption. High-temperature local annealing and ion implantation have demonstrated low loss and high stability. Owing to their thermally driven material modification process, they are most widely used on platforms such as silicon or $Si_3N_4$.

FIB deposition of amorphous carbon offers a room-temperature trimming method, with the potential to be applied across different material platforms. This technique utilizes a focused ion beam to locally decompose precursor gas molecules, enabling precise deposition of diamond-like amorphous carbon[32–34]. Over the past few decades, the technique has been mainly used for the fabrication of 3D nanostructures[30,35,36], though diamond-like amorphous carbon can also be used in anti-reflection coating[37–39], protective coatings[40,41], and as hard masks for etching[42,43]. However, its application in post-fabrication trimming of PICs has not yet been explored. Given that FIB technology is well-established and widely available in research facilities, this presents a valuable opportunity for precise post-fabrication trimming in PICs.

To validate the trimming capability of FIB carbon deposition, we investigated asymmetric directional couplers (A-DCs) as a representative device. A-DCs are widely used for on-chip mode conversion[44–46] and rely on precise phase matching between selected modes, which occurs when their effective refractive indices ($n_{eff}$) are equal at a given wavelength. Because of this strong dependence on precise phase matching, A-DCs are sensitive to fabrication imperfections, making them an ideal platform for evaluating post-fabrication trimming techniques.

In this study, we demonstrate FIB carbon deposition as a localized trimming technique for PICs. To validate its effectiveness, we employ A-DCs as a sensitive test structure, where trimming allows precise control of device responses. We also quantify the trimming-induced insertion loss, thermal behavior, accessible tuning range, and stability over a two-month observation period, demonstrating the potential of this approach for high-resolution trimming in representative PIC devices.

## Results

To visualize the trimming process, consider an A-DC composed of a narrow waveguide (width $w_0$) placed adjacent to a wider waveguide (width $w_1$), separated by a gap $g$. Both waveguides share the same height $h$ and sit atop a buried oxide (BOX) layer on a silicon substrate, and are surrounded by air. Trimming begins when a carbon-containing precursor gas is directed toward the narrow waveguide through a gas nozzle, as shown in Fig. 1a. The $Ga^+$ ion beam induces local decomposition of the precursor gas molecules when exposed to the waveguide, resulting in the accumulation of carbon in the irradiation region with nanometer-scale precision (Supplementary Fig. 1 for the actual FIB sample stage diagram). In this study, we chose to deposit carbon on the narrow waveguide to control the $n_{eff}$ of the $TE_0$ mode for demonstration purposes. Carbon can also be deposited on the wide waveguide to control higher-order modes[31].

Complementing the schematic, a fabricated A-DC based on the $TE_2$ mode (Supplementary Fig. 2 for fabrication process) is shown in Fig. 1b. Light is input via a grating coupler and is equally split by a multimode interferometer. Half of the input light is directed to the output grating coupler (indicated by the white dashed box) and measured by a photodetector. The resulting transmission spectrum is used as the reference for normalization. The first A-DC converts the remaining light from the $TE_0$ mode to the $TE_2$ mode. After conversion, the light propagates along the bus waveguide in the $TE_2$ mode and is converted back to the $TE_0$ mode by the second A-DC before reaching the output grating coupler. For simplicity, we assume in this study that the measured insertion loss is dominated by the mode conversions.

Building on the device structure, we simulated the mode conversion behavior of A-DCs based on the $TE_1$ and $TE_2$ modes, using

parameters of $w_0 = 1200$ nm, $g = 300$ nm, and $h = 335$ nm. By adjusting the bus waveguide width, an insertion loss of 0.055 dB for the $TE_1$ mode and 0.045 dB for the $TE_2$ mode is achieved for the A-DC at 1550 nm, as shown in Fig. 1c and f. The insets show that the $TE_0$ mode light in the narrow waveguide is converted into the higher-order mode light in the wide waveguide. Figure 1d and g exhibit the transmission of the A-DCs at a wavelength of 1550 nm under different $w_1$ values, where the highest transmission appears at $w_1 = 2550$ nm and 3900 nm, respectively. These structures were then fabricated for verification, showing only minimal deviation from the expected waveguide widths, primarily due to resist shrinkage during the hot-plate baking process. We observe the measured insertion losses to be 0.4 dB and 0.5 dB with waveguide widths of 2475 nm and 3750 nm for $TE_1$ and $TE_2$ modes, in that order. In addition, the observed insertion losses of 2.7 dB and 7.4 dB for the $TE_1$ and $TE_2$ modes at $w_1 = 2550$ nm and 3900 nm, respectively, further highlight the necessity of a wide tuning range for mode control (Fig. 1e and h).

In the experiment, the $w_1$ value that achieved the highest transmission was slightly lower than the simulation results for the two modes. This is because resist shrinkage has a greater influence on the $n_{eff}$ of the $TE_0$ mode in the narrow waveguide, which can be compensated by depositing carbon. Figure 2a shows the $n_{eff}$ of the $TE_0$ mode can be flexibly controlled by changing the carbon geometry. The refractive index $n$ and extinction coefficient $k$ of the carbon in our simulation models are 2.48 and 0.019, respectively[47]. Due to its small geometric dimensions and moderate refractive index, the deposited carbon has a negligible effect on the field distribution of the $TE_0$ mode, as shown in Fig. 2b. Similar outcomes are observed for waveguides with silica cladding as well, confirming applicability beyond air (Supplementary Fig. 3). To study the effect of carbon on the optical response of A-DCs, we reduce $w_0$ from 1200 nm to 1150 nm while keeping $w_1$ unchanged, thereby breaking the phase-matching conditions. As shown in Fig. 2c and d, after placing the carbon model on the narrow waveguide, the insertion loss decreases from approximately 5 dB to 0.24 dB ($TE_1$ mode) and 0.33 dB ($TE_2$ mode).

The trimming-induced insertion losses are approximately 0.2 dB for the $TE_1$-mode A-DCs and 0.24 dB for the $TE_2$-mode A-DCs, as the deposited carbon is not perfectly lossless at 1550 nm. However, a small $k$ has only a minimal impact on the overall insertion loss. As shown in Fig. 2e, when the $k$ value gradually increases from 0 to 0.1, the insertion loss of the $TE_1$-mode A-DC increases from 0.035 dB to 1.08 dB. This can be generalized as an approximation: for every 0.01 increase in the $k$ value here, the insertion loss increases by ~0.1 dB. Since the output of the bar port is almost the same in Fig. 2f, the increased insertion loss is mainly caused by the increase in light absorption. The $TE_2$ mode-based A-DC also exhibits the same trend, but with an insertion loss that increases by ~0.13 dB for every 0.01 increase in the $k$ value (Fig. 2g and h). This is mainly owed to the crossover length of the $TE_2$ mode, 110 µm, which is longer than the 90 µm in the $TE_1$ mode.

To explore the impact of carbon deposition at the nanoscale, electron energy loss spectroscopy (EELS) spectra were first used to determine the bandgaps of the SiN waveguide (5.25 eV) and the deposited carbon (3.9 eV), the latter corresponding to diamond-like carbon[48]. As a complementary technique, scanning transmission electron microscopy (STEM) imaging with energy-dispersive spectroscopy (EDS) analysis was used to examine the cross-section of the cut narrow waveguide and reveal how the carbon integrates with the underlying structure. Figure 3a and b show the elemental mapping of the waveguide after carbon deposition. In the elemental maps of carbon and gallium, we observe an increase in brightness in the local area above the waveguide, indicating that the deposited layer has a mixture of gallium and carbon[49]. The gallium originates from the $Ga^+$ ion beam used during the FIB-assisted carbon deposition and, importantly, is confined to the deposited layer and absent from the underlying SiN waveguide. This is consistent with the use of a low probe current (20

pA) at 30 kV acceleration voltage, conditions that favor surface-assisted carbon deposition rather than gallium implantation. Under these parameters, the interaction volume of Ga⁺ ions is minimal relative to the waveguide core, keeping the process firmly in the deposition regime.

This is further supported by the line scan profile in Fig. 3c, showing a localized increase in both carbon and gallium above the waveguide. According to the elemental maps, the carbon deposition is limited to the designated area, with no significant changes in elemental composition detected in the underlying SiN. Furthermore, the plasmon map in Fig. 3d and the corresponding high-angle annular dark-field (HAADF) micrograph in Fig. 3e together corroborate this conclusion: the plasmon map reveals negligible variations in the plasmon energy and intensity after carbon deposition, suggesting that the collective electronic response of the material remains unchanged. Combined with the HAADF micrograph, these observations confirm that

the deposition is confined to this region and no significant structural alterations are detectable within the resolution limits of this method. Therefore, this method is expected to be compatible with different platforms without requiring any platform-specific modifications.

Since the carbon is not completely lossless, we prepared simple waveguide transmission test structures and Mach-Zehnder interferometers (MZIs) (Supplementary Fig. 4 and Fig. 5) to quantify the insertion loss per unit length and the corresponding change in effective refractive index $\Delta n_{\mathrm{eff}}$, respectively. The deposited carbon exhibits a unit-length loss of 76 dB cm⁻¹, as shown in Fig. 4a. Under the same deposition conditions, the induced $\Delta n_{\mathrm{eff}}$ is found to be approximately 0.013. Following the characterization of the loss and $\Delta n_{\mathrm{eff}}$, we further examined how these parameters evolve after thermal treatment at different temperatures. The thermal treatment was performed in air using a heating-plate ramp rate of 15 °C min⁻¹, followed by a 20 min hold at the target temperature and subsequent cooling to room tem-

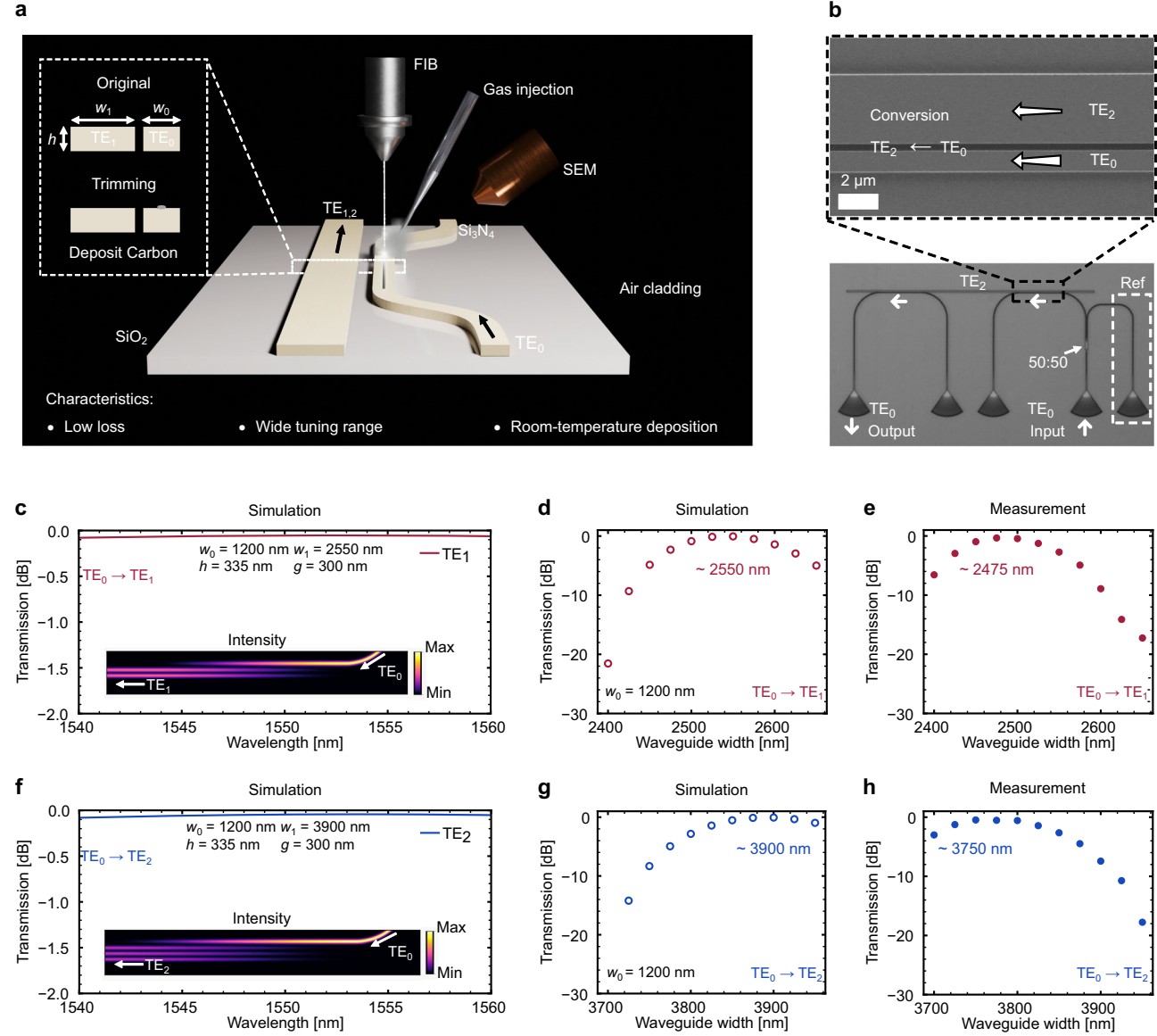

**Fig. 1 | Schematic diagram of a trimmed asymmetric directional coupler (A-DC) using carbon deposition, and the transmission response of the untrimmed A-DCs. a** Schematic of a trimmed A-DC by carbon deposition. **b** Images of a fabricated A-DC for mode conversion between TE₀ and TE₂ modes. **c** Simulated transmission spectrum and intensity distribution of the TE₁-mode A-DC. **d**, **e** Simulated and measured transmission of the TE₁-mode A-DCs with increasing $w_1$. The difference between the simulation results and the measurement results is due to fabrication errors in the width of the waveguides. **f** Simulated transmission spectrum and intensity distribution of the TE₂-mode A-DC. **g**, **h** Simulated and measured transmission of the TE₂-mode A-DCs with increasing $w_1$.

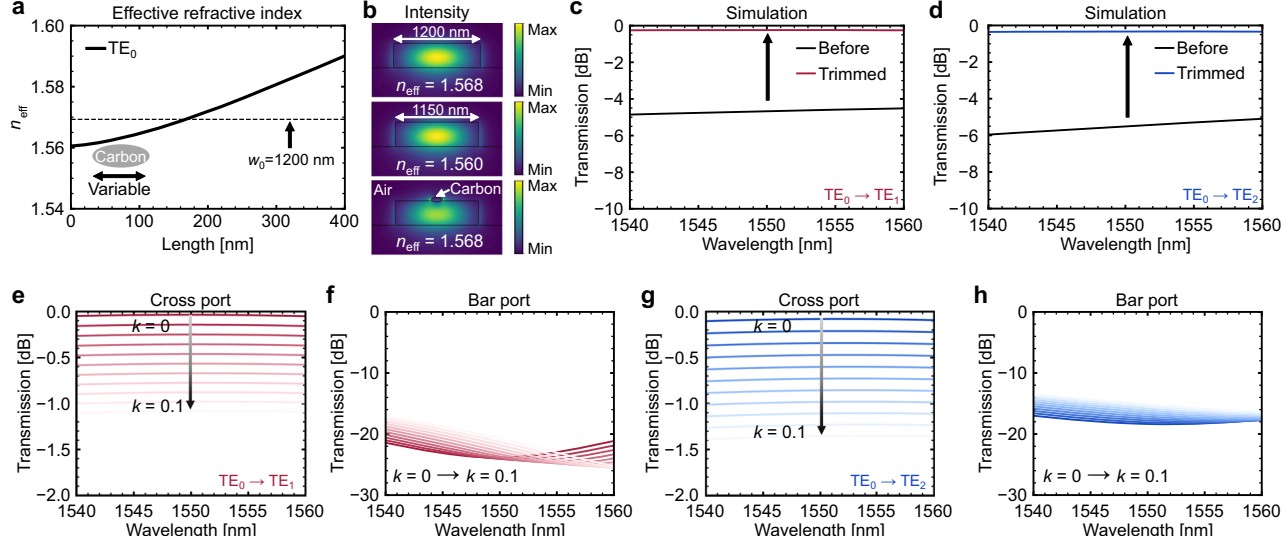

**Fig. 2 | Simulation results of asymmetric directional couplers before and after carbon deposition. a** Effective refractive index $n_{eff}$ of the $TE_0$ mode varies with the major axis length of carbon. When $w_0 = 1150$ nm and major axis length = 160 nm, the $n_{eff}$ of the $TE_0$ mode is equal to the $n_{eff}$ at $w_0 = 1200$ nm. **b** Intensity distribution of a 1200 nm-wide waveguide, a 1150 nm-wide waveguide, and a 1150 nm-wide waveguide with carbon deposition. The carbon is elliptical with major axis and minor axis lengths of 160 nm and 60 nm. The refractive index $n$ and extinction coefficient $k$ of the deposited carbon are 2.48 and 0.019[47], respectively. Transmission spectra of A-DC with 1150 nm-wide waveguide before and after depositing carbon for **c** $TE_1$ mode and **d** $TE_2$ mode. **e, f** When the $k$ of the deposited carbon is assumed to vary from 0 to 0.1, the transmission spectra of the A-DC based on the $TE_1$ mode. **g, h** Assuming that the $k$ of the deposited carbon varies from 0 to 0.1, the transmission spectra of the A-DC based on the $TE_2$ mode.

perature (RT). Measurements were initiated immediately after the sample cooled and were completed within 20 min. Insertion loss remains essentially unchanged below 200 °C, begins to increase at 250 °C, and reaches ~1.6 times its initial value after heat treatment at 350 °C. Correspondingly, the $\Delta n_{eff}$ gradually decreases with temperature and retains nearly half of its initial value after the high-temperature thermal treatment, as shown in Fig. 4b and c. These thermal trends indicate that the trimming performance remains stable up to a temperature around 250 °C. Above this point, the deposited carbon begins to gradually degrade toward a more graphitic form, placing the upper limit of the thermal treatment temperature in the 250 – 300 °C range[33,50,51].

With both the elemental distribution and optical loss confirmed, attention turns to the trimmed region within the A-DC structures. A scanning electron microscope (SEM) image of the $TE_1$-mode A-DC after carbon deposition is presented in Fig. 4d. A dimmer line can be observed on the narrow waveguide, formed by carbon deposition, which increases the $n_{eff}$ value of the $TE_0$ mode to meet the phase-matching conditions. The length of the deposited carbon is designed to match the crossover length of the A-DC, ensuring efficient mode conversion across the entire coupling region. After carbon deposition, the measured insertion loss of the A-DC is reduced from 1.81 dB to 0.36 dB, as shown in Fig. 4e. To corroborate the experimental results, we conducted simulations, and the results in Fig. 4f exhibit good agreement with measurement results, further validating the effectiveness of carbon deposition in controlling the mode conversion. In the SEM image of the $TE_2$-mode A-DC after carbon deposition (Fig. 4g), we observe that the carbon is not perfectly centered in the narrow waveguide; however, according to our simulation results even when the center position offset reaches ±100 nm, there is almost no influence on the $n_{eff}$ of the $TE_0$ mode. Figure 4h shows a significant transmission improvement, where the insertion loss at ~1550 nm decreased from 4.5 dB to 0.82 dB after trimming. This result is in close agreement with the simulated data presented in Fig. 4i. Herein, we discuss carbon deposition using $TE_1$ and $TE_2$ modes as examples, but it can be extended to higher-order modes, demonstrating excellent versatility.

Carbon deposition also provides wide tuning capability, ensuring compatibility with a broad range of initial waveguide conditions, as shown in Fig. 5a. Here, the initial insertion losses of the A-DCs to be trimmed are gradually increased from 1.8 dB to 17.8 dB. For the $TE_1$-mode A-DCs at $w_1 = 2525$ nm (marked as 1), the improvement in the insertion loss is as fine as 1.46 dB, from 1.83 dB to 0.37 dB, confirming the feasibility of this method for fine-tuning. For the $TE_2$-mode A-DCs at $w_1 = 3950$ nm (marked as 6), the improvement in the insertion loss can reach as high as 16.1 dB, from 17.8 dB to 1.7 dB.

To evaluate the temporal stability of this non-volatile post-fabrication trimming method, we measured the transmission of the $TE_1$-mode A-DC at different time points. As shown in Fig. 5b, the transmission slightly decreases during the first two weeks and then stabilizes. The minor change in transmission observed within the first two weeks is likely attributed to stress relaxation or gradual structural stabilization of the deposited carbon, a process can be accelerated by thermal treatment. To verify this hypothesis, a device was prepared and heated at 200 °C, followed by measurement at different time intervals. The heating temperature was selected in accordance with our thermal treatment study to prevent noticeable performance degradation. The optical response of the device tended to stabilize after approximately 30 min of thermal treatment (Supplementary Fig. 6 and Fig. 7). Based on the results, the chip containing the $TE_2$-mode A-DC was placed on a hot plate at 200 °C for 30 min. As shown in Fig. 5c, the transmission exhibited a slight decrease after heating, but it remains stable over the following weeks after the heat treatment.

Due to its mask-free nature, the FIB-based carbon deposition demonstrates exceptional flexibility in the post-fabrication trimming. This trimming method, combined with A-DCs can be applied to photonic crossbar arrays[9,52,53], which are commonly used to perform matrix-vector multiplication (MVM) operations to accelerate artificial intelligence tasks. As shown in Fig. 6a, the A-DCs (indicated by a white dashed box) can be added before the output grating coupler to combine the output light of two small 9 × 2 crossbar arrays, which is equivalent to performing an addition operation (Supplementary Fig. 8 for device layout).

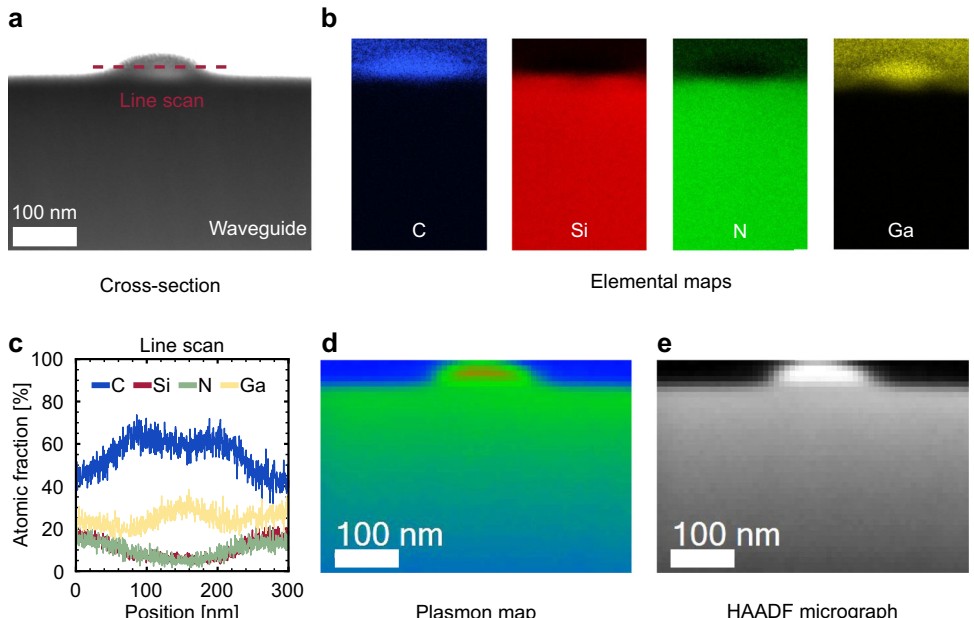

**Fig. 3 | Transmission electron microscope study of the deposited carbon.** **a** Cross-section of the cut waveguide with carbon deposition. **b** STEM-EDS elemental mapping of the waveguide. **c** Line scan profile of the deposited carbon indicated using a red dashed line in the cross-section image. **d**, **e** Plasmon map and HAADF micrograph of the cross-sectional region near the deposited carbon.

Figure 6b shows an image of the A-DC in the very same crossbar configuration. For directional couplers based on the $TE_0$ mode, we observe that after combination, the output power of each input is reduced by 3 dB due to the 50:50 power splitting. Because the orthogonal modes (e.g., $TE_0$ and $TE_1$ modes) are independent, when using an A-DC based on mode-division multiplexing (MDM) technology for combination, the loss here can be approximated as the insertion loss, which is much lower than 3 dB after carbon deposition. Figure 6c shows a comparison of transmitted light before and after carbon deposition. The insertion loss for the conversion from the $TE_0$ mode to $TE_1$ mode decreases from approximately 3 dB to 0.5 dB after trimming, ensuring efficient combination.

The A-DC is used as a demonstrator of the trimming concept; the method itself is not restricted to this particular geometry. We also applied the method to a directional coupler and an MZI, both exhibiting the expected response after carbon deposition (Supplementary Fig. 9). With these validations, we next benchmark the trimming performance of our method using the MZI results. The interference-fringe shift provides a direct and quantitative measure of the induced phase change, enabling a fair comparison with representative non-volatile trimming technologies. Table 1 summarizes selected approaches reported in the literature. Methods based on permanent modifications in the waveguide core material, such as ion implantation and laser-based trimming, have already achieved very low insertion-loss values of ~0.2−0.3 dB per π phase shift[23,54,55]. In addition, a nearly lossless trimming method has recently been demonstrated using high-temperature annealing of undercut waveguides[27]. PCM-based trimming approaches provide large refractive-index modulation through controlled phase transitions and are widely explored for trimming, with recent demonstrations reporting insertion loss of ~0.3−0.4 dB per π phase shift[20,22]. The insertion loss achieved by our method lies within this performance range, supporting its suitability as a device-level trimming option.

## Discussion

We present a trimming method for PICs based on FIB carbon deposition, which is non-volatile, low-loss, deposited at room-temperature, and stable over a two-month observation period. As a representative case, we demonstrate its application in post-fabrication tuning of A-DCs, achieving discrete trimming levels ranging from ~1.5 dB to 16.1 dB. The unit-length excess loss of the deposited carbon is 76 dB cm$^{-1}$, and the additional insertion loss introduced during the A-DC trimming is estimated to be ~0.3 dB, indicating the minimal optical penalty associated with this technique. These results demonstrate the method's feasibility on a fabrication-sensitive component. Furthermore, the trimming does not rely on localized heating, enabling compact layouts and facilitating dense integration. Because the deposition is performed at room temperature and confined to the waveguide surfaces, leaving the waveguide core unaltered, the method can in principle, be applied to both passive and active photonic components, provided the trimmed sections are placed outside the electrically active regions. However, as a FIB-based technique in its present implementation, the proposed approach constitutes a research-stage post-fabrication trimming technique primarily suited for device-level calibration in academic and exploratory fabrication environments.

Crucially, elemental mapping and EELS plasmon map confirm localized surface deposition in the demonstrated structures. Although Ga ions are detected in the STEM EDS analysis, our EELS measurements show that the deposited carbon exhibits a bandgap of 3.9 eV, well within the accepted range for diamond-like amorphous carbon. This, in turn, indicates that Ga content remains below levels that would compromise optical performance, as further supported by the transmission spectra of photonic devices after carbon deposition. Building on these advantages, FIB carbon deposition provides a mask-free trimming approach for the demonstrated SiN-based platform, and has potentials to be extended to other material systems subject to further verification. Together with the device-level optical measurements, these microstructural insights provide a consistent physical picture of the deposition process and provide the foundation for further studies in relation to more complex photonic devices.

## Methods

### Focused ion beam-induced carbon deposition on SiN

Carbon deposition on SiN substrates was performed using a ZEISS Crossbeam 540 FIB/SEM system equipped with a multi-channel Gas Injection System (GIS). Samples were mounted onto the stage and

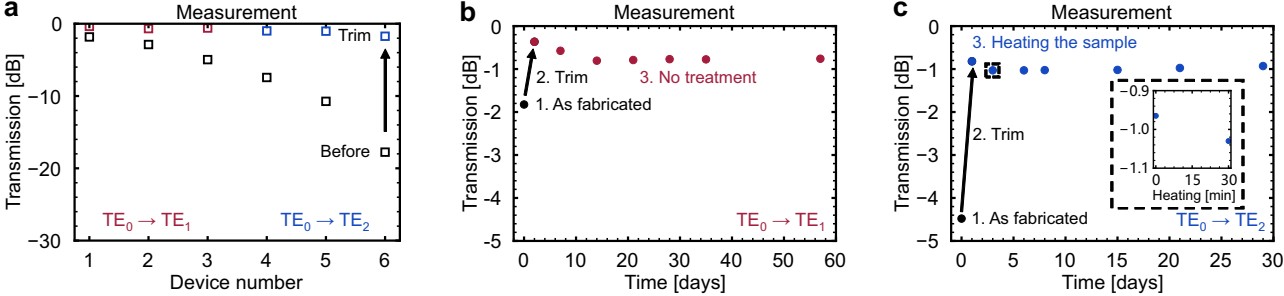

**Fig. 4 | Optical properties of the deposited carbon, post-deposition SEM image of the A-DCs, and their corresponding transmission spectra. a** Insertion loss of deposited carbon at different lengths. **b** Insertion loss measured from multiple devices after thermal treatments at different temperatures. **c** Evolution of $\Delta n_{eff}$ extracted from trimmed devices subjected to thermal treatments at the same set of temperatures. **d** SEM image of the trimmed A-DC for TE$_1$ mode. **e** Measured transmission spectra of the TE$_1$-mode A-DC. **f** Simulation results used to study the measurement results. In this model, $w_0 = 1155$ nm, $w_1 = 2510$ nm, $h = 340$ nm, and $g = 300$ nm. The major and minor axis lengths of the carbon structure are 120 nm and 60 nm, respectively. **g** SEM image of the TE$_2$-mode device. **h** Measured transmission spectra of the TE$_2$-mode A-DC. **i** Simulation results for studying the measured spectra. In this model, $w_0 = 1155$ nm, $w_1 = 3870$ nm, $h = 340$ nm, and $g = 280$ nm. The major and minor axis lengths are 100 nm and 60 nm, respectively.

**Fig. 5 | Transmission of A-DCs at 1550 nm with different initial insertion losses and stability of the optical response of the trimmed A-DCs over time. a** Devices 1–3 are TE$_1$-mode A-DCs with $w_1 = 2525$ nm, 2550 nm, and 2575 nm, respectively. Devices 4–6 are TE$_2$-mode A-DCs with $w_1 = 3900$ nm, 3925 nm, 3950 nm, respectively. The initial insertion losses of these devices vary from 1.8 dB to 17.8 dB. A single carbon deposition can improve insertion loss by up to 16.1 dB. **b** Transmission at 1550 nm for the TE$_1$-mode A-DC over time. The decrease in transmission stops after the second week, and the transmission remains stable for at least 6 weeks thereafter. The final insertion loss is approximately 0.75 dB. **c** Transmission of the A-DC based on the TE$_2$ mode at 1550 nm over time. The device was heated on a hot plate at 200 °C for 30 min. After thermal treatment, the transmission decreases slightly and remains stable for at least 4 weeks thereafter. All the samples in this study were stored in ambient air without any special treatment.

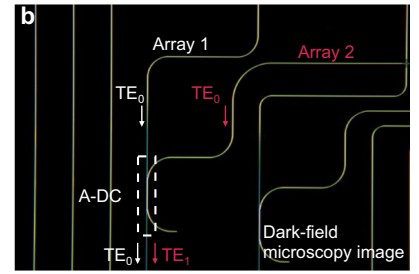
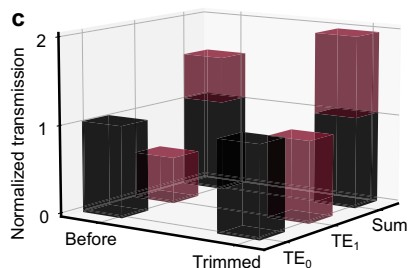

**Fig. 6 | Flexible post-fabrication trimming using carbon deposition in a photonic crossbar array. a** An image of the fabricated photonic crossbar array based on the wavelength-division multiplexing and mode-division multiplexing (MDM) techniques. **b** Schematic of the application of MDM in the photonic crossbar array. **c** Sum of the output powers from the two constituent arrays before and after carbon deposition. Since the $TE_0$ mode light from array 1 experiences almost no loss when passing through the A-DC, its output power here is used as a normalized reference.

**Table 1 | Comparison of state-of-the-art non-volatile trimming techniques**

| Approach | Platform | $\lambda$ [µm] | $IL_\pi$ [dB] | $L_\pi$ [µm] | $T_{processing}$ [°C] | Ref. |
|---|---|---|---|---|---|---|
| Ion implantation | Si | 1.55 | ~0.2[a] | ~6[a] | ~450 | 55,56 |
| UV laser trimming | SiN | 1.95 | ~0.28[a] | 53[b] | N.A. | 23 |
| fs laser trimming | Si | 1.55 | ~0.31[a] | N.A. | N.A. | 24 |
| Annealing | Si | 1.55 | ~0 | N.A. | ~1000 | 27 |
| $Ge_2Sb_2Te_5$ | Si | 2.32 | 2.6 | 25 | ~150 | 21 |
| $Sb_2Se_3$ | SiN | 1.55 | 1.15 | 15.5 | ~200 | 57 |
| $Sb_2Se_3$ | Si | 1.55 | 0.38 | ~7 | N.A. | 22 |
| $Sb_2S_3$ | SiN | 1.55 | 0.89 | 38.75 | ~300 | 57 |
| $Sb_2S_3$ | Si | 1.55 | 0.33 | 24 | ~310 | 20 |
| FIB carbon deposition | SiN | 1.55 | 0.35 | 50 | RT | c |

*N.A.* Not available in the original reference.
[a]Estimated based on data reported in the reference.
[b]Total length reported for a π phase shift in the parallel-exposure configuration.
[c]Data obtained in this work.

introduced into the chamber, where the stage was tilted to 54° to align the sample surface normal to the FIB column.

Once thermal and vacuum equilibrium was achieved, a carbon precursor gas was introduced via the GIS. Using a FIB probe current of 20 pA at 30 kV, a carbon line of a specified length was deposited directly on top of the waveguides. The ion beam and GIS were activated simultaneously, and the deposition dose was calibrated to achieve a nominal carbon thickness of ~50 nm.

### STEM analysis of the trimmed SiN waveguides

Post-trimming analysis of the SiN waveguides was performed using a Thermo Fisher Scientific FEI Themis 300 G3 Titan scanning/transmission electron microscope (S/TEM), operated at 300/60 kV. Samples were prepared in cross-sectional orientation using the FIB technique on a ZEISS Crossbeam 540 FIB/SEM system. The specimens were thinned to electron transparency for imaging. Elemental mapping was conducted in STEM mode using energy-dispersive X-ray spectroscopy (EDS). To investigate bandgap variations in the trimmed waveguides, electron energy loss spectroscopy (EELS) was performed in monochromated STEM mode at 60 kV. The analysis focused on the low-loss region of the EELS spectrum, which contains information about interband transitions and plasmon excitations that are sensitive to changes in the electronic structure and optical bandgap.

### Data availability

All data supporting the findings of this study are available in the article and its Supplementary Information. Source Data are provided with this paper. Additional raw data are available from the corresponding authors upon request.

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

## Acknowledgements

The authors gratefully acknowledge Prof. Dr. Rasmus R. Schröder for generously providing the experimental facilities, technical assistance, and invaluable support that were essential for this study. R.X. and Z.Z. gratefully acknowledge the Alexander von Humboldt Foundation for providing postdoctoral fellowships. W.H.P.P. acknowledges support from the Deutsche Forschungsgemeinschaft (DFG, German Research Foundation) under project numbers 390761711 and 390900948. This work was also supported by the European Union's Horizon projects (PHONICS, grant no. 101017237; HYBRAIN, grant no. 101046878; 2DNEURALVISION, grant no. 101119489) and the European Research Council (PICNIC, grant no. 101200429) to W.H.P.P.

## Author contributions

Conceptualization: R.X., S.T., and R.R.S., Simulation: R.X., L. McRae, and L. Meyer, Sample fabrication: R.X., Z.T., A.V., and S.T., Data acquisition and analysis: R.X., Z.T., Z.Z, R.P., X.M., Q.Z., and F.B.P., Visualization: R.X., J.B., J.R., and J.D., Writing: All authors, Supervision: W.H.P.P., S.T., R.R.S., and H.B. All authors have approved the final version of the manuscript.

## Funding

## Competing interests

The authors declare no competing interests.
