## [Transparent Peer Review file · Nature Communications]

Localized Carbon Deposition Enables Non-Invasive Trimming of Photonic Integrated Circuits

Corresponding Author: Professor Wolfram Pernice

Version 0:

Reviewer comments:

Reviewer #1

(Remarks to the Author)

Trimming of photonic waveguides is highly topical at present, and this is an approach I have not seen before, so in that sense it is interesting. However, it is very difficult to determine the performance of the devices due to the way in which the results are presented. The authors repeatedly refer to the insertion loss of individual modes without describing what happens to other modes for the same measurement conditions, do if the insertion loss changes, is this due to a change in mode coupling, mode conversion, or just the loss of the carbon. For example, at the bottom of page 7, they say that "The trimming-induced insertion losses are approximately 0.2 dB for the TE1-mode A-DCs and 0.24 dB for the TE2-mode A-DCs, which are slightly higher than those induced by silicon ion implantation¹⁹ or polymer-based trimming¹⁸, as the deposited carbon is not perfectly lossless at 1550 nm." So they are comparing loss to other techniques without saying what the loss of the carbon actually is. So on page 10 they investigate the loss via MZIs. They quote an insertion loss of 0.14dB for a carbon length of 20microns. However, it is not clear what the interference states of the MZIs were before and after the addition of carbon, so the addition of carbon will cause a phase change that merely changes the operating point of the MZI, so this is an unhelpful explanation. Later in the same paragraph, they talk about an insertion loss of 0.3dB for a 90 micron length.

What we actually need is the excess loss due to the addition of carbon. So for their ADCs, all optical power in all output modes is measured before and after the addition of carbon. Alternatively, why not just do a direct cut back measurement of a waveguide with different lengths of carbon added. This would give the loss per unit length of carbon, for a given waveguide/carbon design. This is after all, what they compare to in numerous of their cited references.

A much more fundamental question is how a device will be trimmed by this technique. The authors have not actually performed any trimming at all. What they have done is determined that carbon changes the loss and effective refractive index of waveguide modes, and then carried out a whole series of simulations for different designs. If they want to claim a trimming technology, then do some actual trimming. The choice of a mode converting coupler is a curious one. Yes it is sensitive to fabrication tolerances, but is also a complex vehicle to demonstrate trimming. Why not use something more simple like a conventional direction coupler, or an MZI and change its operating point by trimming. How would a real device actually be trimmed using this technique? It would need to be measured in real time whilst the carbon was deposited (or measured immediately afterwards), to see if the operation was changed from a tolerance impacted device, back to an optimal device. It is completely unclear whether the technique has the precision to do that. Compare, for example to a paper by Logan et al (<https://doi.org/10.1109/SiPhotonics64386.2025.10984480>), where high precision trimming is demonstrated.

The authors also claim that the technique is non-volatile, but in essence all they have done is annealed a device once and left it for 2 months. Even the anneal temperature is questionable, as they just placed it on a hotplate nominally at 200 degrees, without actually measuring the surface temperature of the device, which could be quite different. A proper temperature study is required to establish the failure point, the change in loss, and the change in refractive index, as carried out by other authors in this field.

The comparison to the state of the art is also problematic. They have made no reference to phase change materials used for trimming of which there are many examples (e.g. <https://www.nature.com/articles/s41377-023-01213-3>), and no reference to UV based trimming of Silicon Nitride waveguides (e.g. <https://opg.optica.org/prj/fulltext.cfm?uri=prj-8-5-677&id=431148>). The latter seems to be much easier to implement than the proposed technique, and equally effective, arguably more effective.

There are also sloppy parts of the paper where they refer to parts of other papers to make a point, but then ignore that technology when it comes to comparisons. For example, when comparing to the state of the art at the beginning of the paper, they make no mention of Germanium implantation into Silicon as an alternative, yet when they are discussing loss at the top of page 8, they suddenly compare to Germanium implanted technology because they can claim their loss is lower. What they don't say about that technology is that it is higher precision, demonstrably non-volatile, and actually of comparable loss per device because a very small length is needed to trim a device. Actual trimming has also been demonstrated in that technology.

In summary, the technology of the paper is interesting but the paper content is poor, lacks experimental evidence of loss, refractive index change and device trimming, or proper justification of non-volatility. These are all key requirements, so this paper is not suitable for publication in its current form. I suggest rejection.

Reviewer #2

(Remarks to the Author)

In the manuscript "Localized Carbon Deposition Enables Non-Invasive Trimming of Photonic Integrated Circuits" by R. Xu et al., the authors propose and demonstrate refractive index trimming of silicon nitride waveguides by focused ion beam carbon deposition. The demonstration of the trimming method is comprehensive and the reviewer believes the results will be of broad interest to the integrated photonics community. However, there are multiple issues that limit the clarity of the results and context. The reviewer recommends the following major revisions for this manuscript:

1. The silicon nitride devices have no top cladding, but this is not clearly stated in the manuscript. This detail is critical and should be mentioned in the text. The illustration in Fig. 1(a) is not sufficient to convey this point since it is very common for device illustrations to focus on waveguide core features and not show cladding. The only direct mention of the device cladding appears to be in Fig. S2 of the supplementary material.
2. Related to comment #1, the trimming method appears to require no top cladding on the waveguides (for carbon deposition directly on the waveguide top surfaces). Top cladding is important for enabling active photonic devices, requiring metal contacts (in addition to protecting the devices from contamination and humidity). Are there any ways this method could be compatible with cladded devices or at least photonic platforms using a cladding on most devices? This topic is not discussed in the manuscript, but is critical to understanding compatible circuits and applications.
3. The abstract states "These results establish FIB carbon deposition as a robust and broadly applicable PIC trimming technique, empowering optical communications, computing, and beyond to meet rising data demands." This claim should be discussed further in the context of whether the trimming technique is compatible with active photonic circuits (which are very often necessary for the applications mentioned).
4. It's common in index trimming reports to characterize the effective index and propagation loss changes in the waveguide. To enable direct comparison with past work, these measurements should be reported.
5. In Supplementary Note S4, it's mentioned that ring resonators were measured with carbon deposition. Through linewidth and resonance wavelength measurements before and after carbon deposition, it should be possible to calculate the change in propagation loss and effective index of the waveguide. It appears possible to extract this information from Fig. S5 and this analysis should be performed.
6. In Fig. 6, trimming of asymmetric directional couplers is performed in a photonic crossbar array circuit. A schematic of the circuit should be provided. The details of this circuit are not clear from Fig. 6(a).
7. When discussing alternative trimming methods in the introduction, the manuscript states "The trimming method based on micro-heaters usually requires continuous heating to maintain the desired tuning states, resulting in significant power consumption and can cause thermal crosstalk." There are multiple reported methods of thermal trimming where devices are locally annealed using micro-heaters and continuous heating is not required. This statement should be revised and additional references added, for example:

Yanran Xie, Henry C. Frankis, Jonathan D. B. Bradley, and Andrew P. Knights, "Post-fabrication resonance trimming of Si₃N₄ photonic circuits via localized thermal annealing of a sputter-deposited SiO₂ cladding," *Opt. Mater. Express* 11, 2401-2412 (2021)

Tianyuan Xue, Hannes Wahn, Andrei Stalmashonak, Joyce K. S. Poon, Wesley D. Sacher, "In situ Thermal Trimming of Waveguides in a Standard Active Silicon Photonics Platform," arXiv:2506.13578 [physics.optics] (2025)

Yating Wu, Haozhe Sun, Bo Xiong, Yalong Yv, Jiale Zhang, Zhaojie Zheng, Wei Ma, Tao Chu, "Lossless, Non-Volatile Post-Fabrication Trimming of PICs via On-Chip High-Temperature Annealing of Undercut Waveguides," arXiv:2506.18633 [physics.optics] (2025)

Version 1:

Reviewer comments:

Reviewer #1

(Remarks to the Author)

The paper is now significantly improved and can be published after a couple of minor further modifications:

Firstly, the authors express the excess loss due to carbon as $0.0076 \text{ dB } \mu\text{m}^{-1}$. It should be expressed in the more conventional units of dB/cm. i.e. 76 dB/cm . I suspect they want to make the loss appear very low, but I find it misleading, and they can easily follow the value expressed in dB/cm with the insertion loss for a very short section of a few tens of microns, which would still be very low.

The authors argue that the method is "well-suited for a wide range of PIC applications demanding precise optical control", and "provides a practical and forward-compatible pathway toward large-scale, energy efficient, and yield-enhancing photonic integration".

However, FIB processing is not really a commercial fabrication process, the use of carbon is problematic as it contaminates fabrication processes, and the method only works for waveguides without cladding, as carbon treatment would preclude these waveguides from going back into a fab for deposition of a cladding. Therefore, I cannot agree that the process is well suited for a wide range of PIC applications, nor that it provides a practical pathway for large scale integration. It does however, suit University fabs, or some research labs that do not have strict contamination policies. Therefore, these phrases should be removed and replaced with more realistic conclusions.

Reviewer #2

(Remarks to the Author)

The authors have addressed all of my comments through additional characterization work and clarifications. I recommend that the manuscript be accepted for publication.

Version 2:

Reviewer comments:

Reviewer #1

(Remarks to the Author)

This is the second review following a previous review and corrections by the authors. The paper is indeed much better, and the authors have addressed some of my concerns. However, I believe they have not addressed all of my concerns. They are still overclaiming the impact of the technology. They have made much of the additional 2 paragraphs showing that the technique is only suitable for research environments, which are welcome, However, in the abstract a sentence remains that says "These results establish FIB carbon deposition as a robust and broadly applicable PIC trimming technique...."

These results do not do that. They show that the technique has potential, but they have not demonstrated that it is either widely applicable nor robust. If it were widely applicable it would be able to be used widely in fabrication facilities, but I doubt any commercial facility will allow wafers with Carbon on the surface to be processed. That means that claddings cannot be deposited, and almost no commercial PICs would be used without a cladding. The authors mention in passing some results with a cladding, but these are not discussed in any meaningful way in the paper. Secondly, I suspect the word "robust" is used because they have monitored the devices over 2 months. This does not mean that the samples are stable over years, as is effectively claimed. They have no idea what would happen under accelerated aging. That does not mean that the work is without promise and value, but it does mean that they should not claim things that have not been demonstrated. This is particularly pertinent because in the second paragraph of the introduction, they criticise PCM based methods by saying ".....although its long-term stability can be limited."

In the next paragraph they claim that FIB technology is widely available in advanced manufacturing laboratories. Which implies it is an established manufacturing technique for PICs. This is not true, and is not compatible with my previous comments about the commercial applicability of the technology, which they say they agree with in the attached letter.

In the second paragraph of page 3 they say "we demonstrate FIB carbon deposition as a non-invasive technique with great stability". As discussed above, they have not demonstrated this. They have merely demonstrated stability in a period between 2 weeks and 2 months.

Then there is a whole discussion about different technologies that they describe as invasive or non invasive, classifying their technology as non invasive, which suggests it is somehow better than what they call invasive technologies/ However, these definitions are not appropriate or correct. The cladding of a waveguide is an integral part of the waveguide structure and their results are based on waveguides for which a cladding cannot be used, except in contaminated fabrication facilities, and even then they have not demonstrated a cladding on top of their carbon deposition, and therefore have not assessed its impact on the integrity of any cladding. Thus I cannot accept that their technique is non invasive, unless they actually demonstrate that. Again they are over-claiming conclusions. These claims are repeated in the discussion section on page 11 ("broadly applicable", "long term stability", "preserves device integrity").

Therefore, I'm afraid the paper is still not suitable for publication until the unjustified claims are removed.

Version 3:

Reviewer comments:

Reviewer #1

(Remarks to the Author)

I am now content that the authors have addressed my concerns and the paper is acceptable for publishing.

Manuscript number: NCOMMS-25-65564

Response to Reviewer 1:

We would like to thank Reviewer 1 for the insightful and detailed comments, which have inspired and guided the design of a broader set of experiments—carefully carried out in accordance with the reviewer's suggestions—and the development of a more in-depth argumentation. These improvements have significantly strengthened the credibility of the manuscript and the supplementary materials.

Below is a point-by-point response detailing how each comment has been addressed. We format the original comments by the referee in green, and show our revision in blue font.

Original comment:

Trimming of photonic waveguides is highly topical at present, and this is an approach I have not seen before, so in that sense it is interesting. However, it is very difficult to determine the performance of the devices due to the way in which the results are presented. The authors repeatedly refer to the insertion loss of individual modes without describing what happens to other modes for the same measurement conditions, do if the insertion loss changes, is this due to a change in mode coupling, mode conversion, or just the loss of the carbon. For example, at the bottom of page 7, they say that "The trimming-induced insertion losses are approximately 0.2 dB for the TE1-mode A-DCs and 0.24 dB for the TE2-mode A-DCs, which are slightly higher than those induced by silicon ion implantation¹⁹ or polymer-based trimming¹⁸, as the deposited carbon is not perfectly lossless at 1550 nm." So they are comparing loss to other techniques without saying what the loss of the carbon actually is. So on page 10 they investigate the loss via MZIs. They quote an insertion loss of 0.14dB for a carbon length of 20microns. However, it is not clear what the interference states of the MZIs were before and after the addition of carbon, so the addition of carbon will cause a phase change that merely changes the operating point of the MZI, so this is an unhelpful explanation. Later in the same paragraph, they talk about an insertion loss of 0.3dB for a 90 micron length. Comment 1

What we actually need is the excess loss due to the addition of carbon. So for their ADCs, all optical power in all output modes is measured before and after the addition of carbon. Alternatively, why not just do a direct cut back measurement of a waveguide with different lengths of carbon added. This would give the loss per unit length of carbon, for a given waveguide/carbon design. This is after all, what they compare to in numerous of their cited references. Comment 2

A much more fundamental question is how a device will be trimmed by this technique. The authors have not actually performed any trimming at all. What they have done is determined that carbon changes the loss and effective refractive index of waveguide modes, and then carried out a whole series of simulations for different designs. If they want to claim a trimming technology, then do some actual trimming. The choice of a mode converting coupler is a curious one. Yes it is sensitive to fabrication tolerances, but is also

a complex vehicle to demonstrate trimming. Why not use something more simple like a conventional direction coupler, or an MZI and change it's operating point by trimming. How would a real device actually be trimmed using this techniques? It would need to be measured in real time whilst the carbon was deposited (or measured immediately afterwards), to see if the operation was changed from a tolerance impacted device, back to an optimal device. It is completely unclear whether the technique has the precision to do that. Compare, for example to a paper by Logan et al (<https://doi.org/10.1109/SiPhotonics64386.2025.10984480>), where high precision trimming is demonstrated. Comment 3

The authors also claim that the technique is non-volatile, but in essence all they have done is annealed a device once and left it for 2 months. Even the anneal temperature is questionable, as they just placed it on a hotplate nominally at 200 degrees, without actually measuring the surface temperature of the device, which could be quite different. A proper temperature study is required to establish the failure point, the change in loss, and the change in refractive index, as carried out by other authors in this field. Comment 4

The comparison to the state of the art is also problematic. They have made no reference to phase change materials used for trimming of which there are many examples (e.g. <https://www.nature.com/articles/s41377-023-01213-3>), and no reference to UV based trimming of Silicon Nitride waveguides (e.g. <https://opg.optica.org/prj/fulltext.cfm?uri=prj-8-5-677&id=431148>). The latter seems to be much easier to implement than the proposed technique, and equally effective, arguably more effective. Comment 5

There are also sloppy parts of the paper where they refer to parts of other papers to make a point, but then ignore that technology when it comes to comparisons. For example, when comparing to the state of the art at the beginning of the paper, they make no mention of Germanium implantation into Silicon as an alternative, yet when they are discussing loss at the top of page 8, they suddenly compare to Germanium implanted technology because they can claim their loss is lower. What they don't say about that technology is that it is higher precision, demonstrably non-volatile, and actually of comparable loss per device because a very small length is needed to trim a device. Actual trimming has also been demonstrated in that technology. Comment 6

In summary, the technology of the paper is interesting but the paper content is poor, lacks experimental evidence of loss, refractive index change and device trimming, or proper justification of non-volatility. These are all key requirements, so this paper is not suitable for publication in its current form. I suggest rejection.

Comment 1:

Trimming of photonic waveguides is highly topical at present, and this is an approach I have not seen before, so in that sense it is interesting. However, it is very difficult to determine the performance of the devices due to the way in which the results are presented. The authors repeatedly refer to the insertion loss of individual modes without describing

what happens to other modes for the same measurement conditions, do if the insertion loss changes, is this due to a change in mode coupling, mode conversion, or just the loss of the carbon. For example, at the bottom of page 7, they say that "The trimming-induced insertion losses are approximately 0.2 dB for the TE1-mode A-DCs and 0.24 dB for the TE2-mode A-DCs, which are slightly higher than those induced by silicon ion implantation¹⁹ or polymer-based trimming¹⁸, as the deposited carbon is not perfectly lossless at 1550 nm." So they are comparing loss to other techniques without saying what the loss of the carbon actually is. So on page 10 they investigate the loss via MZIs. They quote an insertion loss of 0.14dB for a carbon length of 20microns. However, it is not clear what the interference states of the MZIs were before and after the addition of carbon, so the addition of carbon will cause a phase change that merely changes the operating point of the MZI, so this is an unhelpful explanation. Later in the same paragraph, they talk about an insertion loss of 0.3dB for a 90 micron length.

Our response:

We sincerely appreciate your recognition of the innovative nature of our technology. We fully acknowledge the importance of including data that enable a direct comparison with other non-volatile trimming techniques. Following this valuable suggestion, we have conducted the corresponding loss analysis and incorporated the results into our response to comment 2.

We previously estimated the excess loss introduced by the deposited carbon from the extinction ratio (ER) of the MZI. As shown in Figure RL1, carbon deposition not only shifts the interference fringes but also increases the loss in one arm of the MZI. This additional loss alters the intensity balance between the two arms, thereby modifying the ER. By comparing the ERs before and after deposition, we can extract the attenuation of the carbon-deposited arm and subsequently calculate the corresponding insertion loss. The relation between the ER of the MZI and the power ratio α between the two arms is given by

$$ER = 10 \log_{10} \left(\frac{I_{max}}{I_{min}} \right) = 10 \log_{10} \left(\frac{1 + \sqrt{\alpha}}{1 - \sqrt{\alpha}} \right)^2$$

where I_{max} and I_{min} denote the maximum and minimum transmission of the interference pattern.

Figure RL1: Optical microscope image of a fabricated MZI and its transmission spectra before and after FIB carbon deposition.

Comment 2:

What we actually need is the excess loss due to the addition of carbon. So for their ADCs, all optical power in all output modes is measured before and after the addition of carbon. Alternatively, why not just do a direct cut back measurement of a waveguide with different lengths of carbon added. This would give the loss per unit length of carbon, for a given waveguide/carbon design. This is after all, what they compare to in numerous of their cited references.

Our response:

We sincerely thank the reviewer for this valuable and constructive comment, which is highly relevant to improving the completeness of our study. Following your suggestion, we performed additional experiments by depositing carbon with lengths of 50 μm , 100 μm , 150 μm , and 200 μm to determine the loss per unit length. The corresponding results and discussions have been added to the revised manuscript, as detailed in the section below.

Revision:

Main text (last paragraph on page 7):

Since the carbon is not completely lossless, we prepared simple waveguide transmission test structures and Mach-Zehnder interferometers (MZIs) (Supplementary Fig. 4 and Fig. 5) to quantify the insertion loss per unit length and the corresponding change in effective refractive index Δn_{eff} , respectively. The deposited carbon exhibits a unit-length loss of 0.0076 dB μm^{-1} , as shown in Fig. 4a. Under the same deposition conditions, the induced Δn_{eff} is found to be approximately 0.013. Following the characterization of the loss and Δn_{eff} , we further examined how these parameters evolve after thermal treatment at different temperatures. The thermal treatment was performed in air using a heating-plate ramp rate of 15 $^{\circ}\text{C min}^{-1}$, followed by a 20-minute hold at the target temperature and subsequent cooling to room temperature (RT). Measurements were initiated immediately after the sample cooled and were completed within 20 minutes. Insertion loss remains essentially unchanged below 200 $^{\circ}\text{C}$, begins to increase at 250 $^{\circ}\text{C}$, and reaches approximately 1.6 times its initial value after heat treatment at 350 $^{\circ}\text{C}$. Correspondingly, the Δn_{eff} gradually decreases with temperature and retains nearly half of its initial value after the high-temperature thermal treatment, as shown in Fig. 4b and 4c. These thermal trends indicate that the trimming performance remains stable up to a temperature around 250 $^{\circ}\text{C}$. Above this point, the deposited carbon begins to gradually degrade toward a more graphitic form, placing the upper limit of the thermal treatment temperature in the 250–300 $^{\circ}\text{C}$ range^{33,50,51}.

Fig. 4 Optical properties of the deposited carbon, post-deposition SEM image of the A-DCs, and their corresponding transmission spectra. a Insertion loss of deposited carbon at different lengths. **b** Insertion loss measured from multiple devices after thermal treatments at different temperatures. **c** Evolution of Δn_{eff} extracted from trimmed devices subjected to thermal treatments at the same set of temperatures. **d** SEM image of the trimmed A-DC for TE_1 mode. **e** Measured transmission spectra of the TE_1 -mode A-DC. **f** simulation results used to study the measurement results. In this model, $w_0 = 1155$ nm, $w_1 = 2510$ nm, $h = 340$ nm, and $g = 300$ nm. The major and minor axis lengths of the carbon structure are 120 nm and 60 nm, respectively. **g** SEM image of the TE_2 -mode device. **h** Measured transmission spectra of the TE_2 -mode A-DC. **i** Simulation results for studying the measured spectra. In this model, $w_0 = 1155$ nm, $w_1 = 3870$ nm, $h = 340$ nm, and $g = 280$ nm. The major and minor axis lengths are 100 nm and 60 nm, respectively.

Supplementary (Section 4, page 3):

S4. Insertion loss and associated effective refractive index change

To evaluate the insertion loss per unit length and the corresponding Δn_{eff} , we fabricated a series of waveguide transmission test structures and Mach-Zehnder interferometers (MZIs), as shown in Supplementary Fig. 4. Carbon regions with lengths ranging from 50 to 200 μm , in steps of 50 μm , were deposited under identical conditions across all devices.

The insertion loss was extracted using the waveguide transmission test structures by first normalizing the output of the trimmed arm to the reference arm within the same device, and then comparing the normalized transmission before and after deposition. The waveguide transmission test structures yield a unit-length loss of 0.0078 dB μm^{-1} , while the MZIs fabricated in the same batch give a similar value of 0.0074 dB μm^{-1} for the same deposition lengths, as shown in Supplementary Fig. 5. The close agreement between these measurements confirms the reliability of the extracted loss, whose averaged value of 0.0076 dB μm^{-1} is reported in the main text. The Δn_{eff} was subsequently determined from the spectral shifts of these MZIs, giving an average value of approximately 0.013.

Supplementary Fig. 4. **a** Optical microscope and SEM views of the waveguide transmission test structure. **b** Optical microscope and SEM views of the MZI used to extract the loss and the effective refractive index change.

Supplementary Fig. 5. **a** Extracted insertion loss for deposited carbon of different lengths on waveguide transmission test structures. **b** Extracted insertion loss obtained from the MZIs. Carbon deposition on both types of structures was carried out under identical deposition conditions.

Comment 3:

A much more fundamental question is how a device will be trimmed by this technique. The authors have not actually performed any trimming at all. What they have done is determined that carbon changes the loss and effective refractive index of waveguide modes, and then carried out a whole series of simulations for different designs. If they want to claim a trimming technology, then do some actual trimming. The choice of a mode converting coupler is a curious one. Yes it is sensitive to fabrication tolerances, but is also a complex vehicle to demonstrate trimming. Why not use something more simple like a conventional direction coupler, or an MZI and change its operating point by trimming. How would a real device actually be trimmed using this technique? It would need to be measured in real time whilst the carbon was deposited (or measured immediately afterwards), to see if the operation was changed from a tolerance impacted device, back to an optimal device. It is completely unclear whether the technique has the precision to do that. Compare, for example to a paper by Logan et al (<https://doi.org/10.1109/SiPhotonics64386.2025.10984480>), where high precision trimming is demonstrated.

Our response:

We sincerely appreciate the reviewer's insightful comment. In our study, the trimming technique was indeed experimentally demonstrated on asymmetric directional couplers (A-DCs). We chose A-DCs as the representative platform because our trimming mechanism operates through modification of the local effective refractive index n_{eff} . The operation of an A-DC relies on the precise matching of the n_{eff} of the two coupled waveguide modes; and therefore, its coupling efficiency is highly sensitive to even minute changes in n_{eff} . The high sensitivity makes the A-DC a particularly suitable structure to reveal the trimming effect with clear experimental observables.

We fully agree with the reviewer that trimming on conventional devices such as directional couplers and MZIs can serve as more straightforward benchmarks for comparison with other non-volatile trimming methods. To further demonstrate the general applicability of our approach, we have additionally performed trimming experiments on a conventional directional coupler and an MZI. The corresponding before-and-after transmission responses of these devices have been included in the Supplementary Information, further confirming that the proposed technique enables precise post-fabrication tuning across different device geometries.

Revision:

Supplementary (Section S7, page 6):

S7. Trimming of directional couplers and Mach-Zehnder interferometers

The proposed trimming method is not limited to the A-DC in the main text; it can also be applied to other commonly used devices, such as directional couplers and MZIs. To illustrate this, we fabricated a directional coupler designed to exhibit a 0.5 coupling ratio. Due to a slightly narrower fabricated gap, the actual coupling was stronger than intended, resulting in a measured ratio of 0.75. To correct this deviation, we deposited an approximately 8- μm -long region of carbon on the waveguide, corresponding to roughly one-third of the effective coupling length, as shown in Supplementary Fig. 9a and 9b. To introduce a sufficiently large effective-index contrast between the two waveguide modes and ensure efficient coupling suppression, carbon was deposited twice at nearly the same location. After trimming, the coupling ratio was reduced from 0.75 to the intended value of 0.5, as illustrated in Supplementary Fig. 9c.

We next applied the method to MZIs, which offer a convenient platform for benchmarking trimming performance because their interference fringes provide a direct measure of phase changes. A 50- μm -long carbon was deposited on one arm of the fabricated MZI, as shown in Supplementary Fig. 9d and 9e. As seen in Supplementary Fig. 9f, the deposition produces a near- π phase shift, while the corresponding insertion loss of 0.35 dB is extracted from the extinction ratio between the maximum and minimum transmission. This yields a trimming efficiency of 0.35 dB π^{-1} , which we benchmark against representative non-volatile trimming methods in Table 1 of the main text.

Supplementary Fig. 9. Application of FIB carbon deposition in directional couplers and MZIs. **a, b** Optical microscope image and SEM image of the trimmed directional coupler with a coupling length of 25 μm . **c** The transmission spectra of the directional coupler before and after FIB carbon deposition. **d, e** Optical microscope image and SEM image of the trimmed MZI with a carbon-deposition length of 50 μm . **f** Transmission spectra of the trimmed MZI after carbon deposition, showing a near- π phase shift.

Comment 4:

The authors also claim that the technique is non-volatile, but in essence all they have done is annealed a device once and left it for 2 months. Even the anneal temperature is questionable, as they just placed it on a hotplate nominally at 200 degrees, without actually measuring the surface temperature of the device, which could be quite different. A proper temperature study is required to establish the failure point, the change in loss, and the change in refractive index, as carried out by other authors in this field.

Our response:

We fully agree with the reviewer that a systematic temperature study to determine the failure point is important for a proposed trimming method. Following this helpful suggestion, we conducted a series of measurements on devices thermally treated at different temperatures up to 350 $^{\circ}\text{C}$ to assess the trimming response.

Revision:

Main text (last paragraph on page 7):

Since the carbon is not completely lossless, we prepared simple waveguide transmission test structures and Mach-Zehnder interferometers (MZIs) (Supplementary Fig. 4 and Fig. 5) to quantify the insertion loss per unit length and the corresponding change in effective refractive index Δn_{eff} , respectively. The deposited carbon exhibits a unit-length loss of 0.0076 dB μm^{-1} , as shown in Fig. 4a. Under the same deposition conditions, the induced Δn_{eff} is found to be approximately 0.013. Following the characterization of the loss and Δn_{eff} , we further examined how these parameters evolve after thermal treatment at different temperatures. The thermal treatment was performed in air using a heating-plate ramp rate of 15 $^{\circ}\text{C min}^{-1}$, followed by a 20-minute hold at the target temperature and subsequent

cooling to room temperature (RT). Measurements were initiated immediately after the sample cooled and were completed within 20 minutes. Insertion loss remains essentially unchanged below 200 °C, begins to increase at 250 °C, and reaches approximately 1.6 times its initial value after heat treatment at 350 °C. Correspondingly, the Δn_{eff} gradually decreases with temperature and retains nearly half of its initial value after the high-temperature thermal treatment, as shown in Fig. 4b and 4c. These thermal trends indicate that the trimming performance remains stable up to a temperature around 250 °C. Above this point, the deposited carbon begins to gradually degrade toward a more graphitic form, placing the upper limit of the thermal treatment temperature in the 250–300 °C range^{33,50,51}.

Fig. 4 Optical properties of the deposited carbon, post-deposition SEM image of the A-DCs, and their corresponding transmission spectra. a Insertion loss of deposited carbon at different lengths. **b** Insertion loss measured from multiple devices after thermal treatments at different temperatures. **c** Evolution of Δn_{eff} extracted from trimmed devices subjected to thermal treatments at the same set of temperatures. **d** SEM image of the trimmed A-DC for TE₁ mode. **e** Measured transmission spectra of the TE₁-mode A-DC. **f** simulation results used to study the measurement results. In this model, $w_0 = 1155$ nm, $w_1 = 2510$ nm, $h = 340$ nm, and $g = 300$ nm. The major and minor axis lengths of the carbon structure are 120 nm and 60 nm, respectively. **g** SEM image of the TE₂-mode device. **h** Measured transmission spectra of the TE₂-mode A-DC. **i** Simulation results for studying the measured spectra. In this model, $w_0 = 1155$ nm, $w_1 = 3870$ nm, $h = 340$ nm, and $g = 280$ nm. The major and minor axis lengths are 100 nm and 60 nm, respectively.

Comment 5:

The comparison to the state of the art is also problematic. They have made no reference to phase change materials used for trimming of which there are many examples (e.g. <https://www.nature.com/articles/s41377-023-01213-3>), and no reference to UV based trimming of Silicon Nitride waveguides (e.g. <https://opg.optica.org/prj/fulltext.cfm?uri=prj-8->

5-677&id=431148). The latter seems to be much easier to implement than the proposed technique, and equally effective, arguably more effective.

Response:

We truly appreciate the reviewer's comment, which highlights the need for a more complete discussion of the state of the art in non-volatile trimming. In response, we have now incorporated a detailed comparison of various advanced technologies in the main text, including both phase-change-material (PCM)-based approaches and UV-laser trimming of SiN waveguides into our main text. These works are indeed important contributions to the field, and we appreciate the reviewer bringing them to our attention. The corresponding revisions have been added below.

Revision:

Main text (last paragraph on page 10):

The A-DC is used as a demonstrator of the trimming concept; the method itself is not restricted to this particular geometry. We also applied the method to a directional coupler and an MZI, both exhibiting the expected response after carbon deposition (Supplementary Fig. 9). With these validations, we next benchmark the trimming performance of our method using the MZI results. The interference-fringe shift provides a direct and quantitative measure of the induced phase change, enabling a fair comparison with representative non-volatile trimming technologies. For clarity, we categorize representative trimming technologies into two groups: invasive and non-invasive, as shown in Table 1. Invasive trimming methods typically induce permanent modifications in the waveguide core material, and the state-of-the-art methods such as ion implantation and laser-based trimming have already achieved very low insertion-loss values of approximately 0.2–0.3 dB per π phase shift^{23,54,55}. In addition, a nearly lossless trimming method has recently been demonstrated using high-temperature annealing of undercut waveguides²⁷. Compared with invasive methods, non-invasive trimming techniques preserve the waveguide core, thereby offering broader platform compatibility. Among these, programmable PCMs offer large refractive-index modulation through controlled phase transitions and are widely explored for trimming, with recent demonstrations reporting insertion loss of approximately 0.3–0.4 dB per π phase shift^{20,22}. The insertion loss achieved by our method lies within this performance range, supporting its suitability as a practical trimming option.

Table 1 Comparison of state-of-the-art non-volatile trimming techniques.

Invasive type						
Approach	Platform	λ [μm]	IL $_{\pi}$ [dB]	L_{π} [μm]	$T_{\text{processing}}$ [$^{\circ}\text{C}$]	Ref.
Ion implantation	Si	1.55	~0.2 ^a	~6 ^a	~450	55,56
UV laser trimming	SiN	1.95	~0.28 ^a	53 ^b	N.A.	23
fs laser trimming	Si	1.55	~0.31 ^a	N.A.	N.A.	24
Annealing	Si	1.55	~0	N.A.	~1000	27
Non-invasive type						
Approach	Platform	λ [μm]	IL $_{\pi}$ [dB]	L_{π} [μm]	$T_{\text{processing}}$ [$^{\circ}\text{C}$]	Ref.

Ge ₂ Sb ₂ Te ₅	Si	2.32	2.6	25	~150	21
Sb ₂ Se ₃	SiN	1.55	1.15	15.5	~200	57
Sb ₂ Se ₃	Si	1.55	0.38	~7	N.A.	22
Sb ₂ S ₃	SiN	1.55	0.89	38.75	~300	57
Sb ₂ S ₃	Si	1.55	0.33	24	~310	20
FIB carbon deposition	SiN	1.55	0.35	50	RT	^c

^a Estimated based on data reported in the reference.

^b Total length reported for a π phase shift in the parallel-exposure configuration.

^c Data obtained in this work.

N.A.: Not available in the original reference.

Reference:

23. De Paoli, G. et al. Laser trimming of the operating wavelength of silicon nitride racetrack resonators. *Photon. Res.* 8, 677 (2020).

55. Milosevic, M. M. et al. Ion Implantation in Silicon for Trimming the Operating Wavelength of Ring Resonators. *IEEE J. Select. Topics Quantum Electron.* 24, 1–7 (2018).

56. Logan, A. M. et al. Towards precise trimming and programming of photonic integrated circuits. in 2025 IEEE Silicon Photonics Conference (SiPhotonics) (IEEE, London, 2025).

Comment 6:

There are also sloppy parts of the paper where they refer to parts of other papers to make a point, but then ignore that technology when it comes to comparisons. For example, when comparing to the state of the art at the beginning of the paper, they make no mention of Germanium implantation into Silicon as an alternative, yet when they are discussing loss at the top of page 8, they suddenly compare to Germanium implanted technology because they can claim their loss is lower. What they don't say about that technology is that it is higher precision, demonstrably non-volatile, and actually of comparable loss per device because a very small length is needed to trim a device. Actual trimming has also been demonstrated in that technology.

Response:

We thank the reviewer for pointing out this important aspect. Germanium ion implantation has indeed demonstrated excellent performance on silicon platform. In crystalline silicon, implantation-induced lattice damage partially amorphizes the material but can be effectively recovered through high-temperature annealing, enabling large tunability with low residual loss.

In contrast, the material used in our work is silicon nitride (SiN), which is amorphous. Based on our previous experience with silicon ion implantation on SiN, implanted ions tend to accumulate within the waveguide core, and the resulting defects cannot be reversed by annealing. Because germanium also absorbs light at 1550 nm, we therefore assumed that germanium ion accumulation in SiN would introduce non-negligible loss.

With the reviewer's insightful comment, we realized that the situation for germanium ion implantation on crystalline silicon is fundamentally different due to the small device footprint and the recoverability of lattice damage. We have now clarified this platform-dependent distinction and fully revised the corresponding discussion in the manuscript.

Revision:

Main text (second paragraph on page 2):

Various post-fabrication trimming techniques have been proposed to address fabrication imperfections. Volatile approaches are typically based on thermal tuning using micro-heaters^{14,15}, with the trimmed state sustained while electrical power is applied. In contrast, non-volatile approaches, such as electron beam exposure of polymer claddings^{16–18}, programmable phase-change materials (PCMs) integrated on waveguides^{19–22}, laser-based trimming^{23,24}, high-temperature local annealing^{25–27}, and focused ion beam (FIB) processing^{15,28–31}, permanently modify the optical response without sustained power consumption, but rely on distinct physical mechanisms and are implemented under different processing conditions. Polymer cladding trimming is a simple and cost-effective method, although its long-term stability can be limited. PCM-based methods are broadly compatible with multiple material platforms, but they generally introduce higher insertion loss due to mode mismatch and intrinsic absorption. High-temperature local annealing and ion implantation have demonstrated low loss and high stability. Owing to their thermally driven material modification process, they are most widely used on platforms such as silicon or Si₃N₄.

Reference:

19. Meng, J. et al. Electrical programmable multilevel nonvolatile photonic random-access memory. *Light Sci Appl* 12, 189 (2023).

We hope to have addressed all concerns of the referee with the aforementioned revisions to our manuscript and trust that she/he will be in favor of publishing our results in *Nature Communications*.

Response to Reviewer 2:

We sincerely thank the reviewer for the thoughtful and constructive evaluation of our work. The insightful comments have greatly helped us enhance the clarity and completeness of the manuscript. We have carefully addressed each point and revised the manuscript accordingly. Details of the revisions are provided below. We format the original comments by the referee in green, and show our revision in blue font.

Original comment:

In the manuscript "Localized Carbon Deposition Enables Non-Invasive Trimming of Photonic Integrated Circuits" by R. Xu et al., the authors propose and demonstrate refractive index trimming of silicon nitride waveguides by focused ion beam carbon deposition. The demonstration of the trimming method is comprehensive and the reviewer believes the results will be of broad interest to the integrated photonics community. However, there are multiple issues that limit the clarity of the results and context. The reviewer recommends the following major revisions for this manuscript:

1. The silicon nitride devices have no top cladding, but this is not clearly stated in the manuscript. This detail is critical and should be mentioned in the text. The illustration in Fig. 1(a) is not sufficient to convey this point since it is very common for device illustrations to focus on waveguide core features and not show cladding. The only direct mention of the device cladding appears to be in Fig. S2 of the supplementary material. Comment 1

2. Related to comment #1, the trimming method appears to require no top cladding on the waveguides (for carbon deposition directly on the waveguide top surfaces). Top cladding is important for enabling active photonic devices, requiring metal contacts (in addition to protecting the devices from contamination and humidity). Are there any ways this method could be compatible with cladded devices or at least photonic platforms using a cladding on most devices? This topic is not discussed in the manuscript, but is critical to understanding compatible circuits and applications. Comment 2

3. The abstract states "These results establish FIB carbon deposition as a robust and broadly applicable PIC trimming technique, empowering optical communications, computing, and beyond to meet rising data demands." This claim should be discussed further in the context of whether the trimming technique is compatible with active photonic circuits (which are very often necessary for the applications mentioned). Comment 3

4. It's common in index trimming reports to characterize the effective index and propagation loss changes in the waveguide. To enable direct comparison with past work, these measurements should be reported. Comment 4

5. In Supplementary Note S4, it's mentioned that ring resonators were measured with carbon deposition. Through linewidth and resonance wavelength measurements before

and after carbon deposition, it should be possible to calculate the change in propagation loss and effective index of the waveguide. It appears possible to extract this information from Fig. S5 and this analysis should be performed. ^{Comment 5}

6. In Fig. 6, trimming of asymmetric directional couplers is performed in a photonic crossbar array circuit. A schematic of the circuit should be provided. The details of this circuit are not clear from Fig. 6(a). ^{Comment 6}

7. When discussing alternative trimming methods in the introduction, the manuscript states "The trimming method based on micro-heaters usually requires continuous heating to maintain the desired tuning states, resulting in significant power consumption and can cause thermal crosstalk." There are multiple reported methods of thermal trimming where devices are locally annealed using micro-heaters and continuous heating is not required. This statement should be revised and additional references added, for example:

Yanran Xie, Henry C. Frankis, Jonathan D. B. Bradley, and Andrew P. Knights, "Post-fabrication resonance trimming of Si₃N₄ photonic circuits via localized thermal annealing of a sputter-deposited SiO₂ cladding," Opt. Mater. Express 11, 2401-2412 (2021)

Tianyuan Xue, Hannes Wahn, Andrei Stalmashonak, Joyce K. S. Poon, Wesley D. Sacher, "In situ Thermal Trimming of Waveguides in a Standard Active Silicon Photonics Platform," arXiv:2506.13578 [physics.optics] (2025)

Yating Wu, Haozhe Sun, Bo Xiong, Yalong Yv, Jiale Zhang, Zhaojie Zheng, Wei Ma, Tao Chu, "Lossless, Non-Volatile Post-Fabrication Trimming of PICs via On-Chip High-Temperature Annealing of Undercut Waveguides," arXiv:2506.18633 [physics.optics] (2025) ^{Comment 7}

Our response to the reviewer's comments shown using underlines is as follows.

Comment 1:

The silicon nitride devices have no top cladding, but this is not clearly stated in the manuscript. This detail is critical and should be mentioned in the text. The illustration in Fig. 1(a) is not sufficient to convey this point since it is very common for device illustrations to focus on waveguide core features and not show cladding. The only direct mention of the device cladding appears to be in Fig. S2 of the supplementary material.

Our response:

Thank you so much for pointing out this important issue. We have incorporated the relevant information into the main text and added explanatory text to Fig. 1a. The corresponding revisions are provided below.

Revision:

Main text (first paragraph in the Results section on page 3):

To visualize the trimming process, consider an A-DC composed of a narrow waveguide (width w_0) placed adjacent to a wider waveguide (width w_1), separated by a gap g . Both waveguides share the same height h and sit atop a buried oxide (BOX) layer on a silicon substrate, and are surrounded by air. Trimming begins when a carbon-containing precursor gas is directed toward the narrow waveguide through a gas nozzle, as shown in Fig. 1a.

Fig. 1 Schematic diagram of a trimmed A-DC using carbon deposition, and the transmission response of the untrimmed A-DCs. a Schematic of a trimmed A-DC by carbon deposition. b Images of a fabricated A-DC for mode conversion between TE₀ and TE₂ modes. c Simulated transmission spectrum and intensity distribution of the TE₁-mode A-DC. d, e Simulated and measured transmission of the TE₁-mode A-DCs with increasing w_1 . The difference between the simulation results and the measurement results is due to fabrication errors in the width of the waveguides. f Simulated transmission spectrum and intensity distribution of the TE₂-mode A-DC. g, h Simulated and measured transmission of the TE₂-mode A-DCs with increasing w_1 .

Comment 2:

Related to comment #1, the trimming method appears to require no top cladding on the waveguides (for carbon deposition directly on the waveguide top surfaces). Top cladding is important for enabling active photonic devices, requiring metal contacts (in addition to

protecting the devices from contamination and humidity). Are there any ways this method could be compatible with cladded devices or at least photonic platforms using a cladding on most devices? This topic is not discussed in the manuscript, but is critical to understanding compatible circuits and applications.

Our response:

We sincerely thank the reviewer for the valuable comment. It is definitely true that the cladding should be considered for a trimming method. We have incorporated the relevant discussion in the supplementary.

Revision:

Main text (first paragraph on page 5):

In the experiment, the w_1 value that achieved the highest transmission was slightly lower than the simulation results for the two modes. This is because resist shrinkage has a greater influence on the n_{eff} of the TE_0 mode in the narrow waveguide, which can be compensated by depositing carbon. Fig. 2a shows the n_{eff} of the TE_0 mode can be flexibly controlled by changing the carbon geometry. The refractive index n and extinction coefficient k of the carbon in our simulation models are 2.48 and 0.019, respectively⁴⁷. Due to its small geometric dimensions and moderate refractive index, the deposited carbon has a negligible effect on the field distribution of the TE_0 mode, as shown in Fig. 2b. Similar outcomes are observed for waveguides with silica cladding as well, confirming applicability beyond air (Supplementary Fig. 3). To study the effect of carbon on the optical response of A-DCs, we reduce w_0 from 1200 nm to 1150 nm while keeping w_1 unchanged, thereby breaking the phase-matching conditions. As shown in Fig. 2c and 2d, after placing the carbon model on the narrow waveguide, the insertion loss decreases from approximately 5 dB to 0.24 dB (TE_1 mode) and 0.33 dB (TE_2 mode).

Supplementary: (on page 3)

S3. FIB carbon deposition for cladded photonic structures

In practical applications, waveguides are often covered by a protective cladding layer to enable robust operation and future integration. It is therefore essential to assess the compatibility of the FIB carbon deposition with cladded structures. Supplementary Fig. 3a compares the effective refractive index change Δn_{eff} of carbon deposition for waveguides with and without a silica cladding. As confirmed in Supplementary Fig. 3b and 3c, the deposited carbon interacts less strongly with the waveguide mode as the cladding thickness increases and the carbon becomes more distant from the waveguide core, leading to a reduced Δn_{eff} . With a cladding thickness of 500 nm, the resulting Δn_{eff} is approximately one-quarter of the uncladded value.

For relatively thin claddings, the deposited carbon volume can be increased to compensate for the weakened interaction and maintain efficient trimming. However, when the cladding thickness approaches ~ 1000 nm, the interaction becomes too weak to enable meaningful tuning. In such cases, the FIB carbon deposition can be combined with FIB milling: Ga^+ ions can create nanoscale trenches within the SiO_2 cladding¹, reducing the

distance between the deposited carbon and the waveguide core and thereby enhancing the interaction.

Supplementary Fig. 3. Effect of carbon deposition for different cladding thicknesses. a Percentage change in n_{eff} for varying cladding thicknesses, normalized to the no-cladding case ($\Delta n_{\text{eff}} = 0.008$) using the same deposition volume. **b, c** Intensity distribution of carbon-trimmed waveguide with cladding thickness of 335 nm and 500 nm, respectively.

Reference:

1. Menard, L. D. & Ramsey, J. M. Fabrication of Sub-5 nm Nanochannels in Insulating Substrates Using Focused Ion Beam Milling. *Nano Lett.* 11, 512–517 (2011).

Comment 3:

The abstract states "These results establish FIB carbon deposition as a robust and broadly applicable PIC trimming technique, empowering optical communications, computing, and beyond to meet rising data demands." This claim should be discussed further in the context of whether the trimming technique is compatible with active photonic circuits (which are very often necessary for the applications mentioned).

Our response:

We appreciate the reviewer for raising this important point. The FIB carbon deposition is carried out entirely at room temperature and affects only the surface region, without altering the waveguide core. In addition, the process is highly localized, allowing trimming to be performed away from the active region. For these reasons, the method is expected to be fully compatible with active photonic circuits. We have added a corresponding clarification in the discussion section of the manuscript.

Revision:

Main text (first paragraph of the discussion section on page 11):

We present a trimming method for PICs based on FIB carbon deposition, which is non-volatile, low-loss, highly stable, room-temperature, and non-invasive. As a representative case, we demonstrate its application in post-fabrication tuning of A-DCs, achieving discrete trimming levels ranging from ~ 1.5 dB to 16.1 dB. The unit-length excess loss of the deposited carbon is $0.0076 \text{ dB } \mu\text{m}^{-1}$, and the additional insertion loss introduced during the A-DC trimming is estimated to be ~ 0.3 dB, indicating the minimal optical penalty associated with this technique. These results validate the method's effectiveness on a fabrication-sensitive component, while its combined advantages render it broadly applicable well beyond A-DCs. The trimmed devices also maintain stable performance over a period exceeding two months, confirming the long-term reliability of the method.

Furthermore, the trimming does not rely on localized heating, enabling compact layouts and facilitating dense integration. Because the deposition is performed at room temperature and occurs only on the waveguide surfaces, leaving the waveguide core unaltered, the method inherently preserves device integrity and is therefore, in principle, compatible with both passive and active photonic components, provided the trimmed sections are placed outside the electrically active region. These features make the method well-suited for a wide range of PIC applications demanding precise optical control, such as MDM components for expanding processing capacity and large-scale photonic accelerators requiring low-loss signal combination.

Comment 4:

It's common in index trimming reports to characterize the effective index and propagation loss changes in the waveguide. To enable direct comparison with past work, these measurements should be reported.

Our response:

We sincerely appreciate this valuable comment. We acknowledge that these parameters are essential for a comprehensive evaluation, and we have now incorporated the corresponding values in the revised manuscript. The relevant revisions are provided below.

Revision:

Main text (first paragraph on page 7):

Since the carbon is not completely lossless, we prepared simple waveguide transmission test structures and Mach-Zehnder interferometers (MZIs) (Supplementary Fig. 4 and Fig. 5) to quantify the insertion loss per unit length and the corresponding change in effective refractive index Δn_{eff} , respectively. The deposited carbon exhibits a unit-length loss of 0.0076 dB μm^{-1} , as shown in Fig. 4a. Under the same deposition conditions, the induced Δn_{eff} is found to be approximately 0.013. Following the characterization of the loss and Δn_{eff} , we further examined how these parameters evolve after thermal treatment at different temperatures. The thermal treatment was performed in air using a heating-plate ramp rate of 15 $^{\circ}\text{C min}^{-1}$, followed by a 20-minute hold at the target temperature and subsequent cooling to room temperature (RT). Measurements were initiated immediately after the sample cooled and were completed within 20 minutes. Insertion loss remains essentially unchanged below 200 $^{\circ}\text{C}$, begins to increase at 250 $^{\circ}\text{C}$, and reaches approximately 1.6 times its initial value after heat treatment at 350 $^{\circ}\text{C}$. Correspondingly, the Δn_{eff} gradually decreases with temperature and retains nearly half of its initial value after the high-temperature thermal treatment, as shown in Fig. 4b and 4c. These thermal trends indicate that the trimming performance remains stable up to a temperature around 250 $^{\circ}\text{C}$. Above this point, the deposited carbon begins to gradually degrade toward a more graphitic form, placing the upper limit of the thermal treatment temperature in the 250–300 $^{\circ}\text{C}$ range^{33,50,51}.

Fig. 4 Optical properties of the deposited carbon, post-deposition SEM image of the A-DCs, and their corresponding transmission spectra. a Insertion loss of deposited carbon at different lengths. **b** Insertion loss measured from multiple devices after thermal treatments at different temperatures. **c** Evolution of Δn_{eff} extracted from trimmed devices subjected to thermal treatments at the same set of temperatures. **d** SEM image of the trimmed A-DC for TE_1 mode. **e** Measured transmission spectra of the TE_1 -mode A-DC. **f** simulation results used to study the measurement results. In this model, $w_0 = 1155$ nm, $w_1 = 2510$ nm, $h = 340$ nm, and $g = 300$ nm. The major and minor axis lengths of the carbon structure are 120 nm and 60 nm, respectively. **g** SEM image of the TE_2 -mode device. **h** Measured transmission spectra of the TE_2 -mode A-DC. **i** Simulation results for studying the measured spectra. In this model, $w_0 = 1155$ nm, $w_1 = 3870$ nm, $h = 340$ nm, and $g = 280$ nm. The major and minor axis lengths are 100 nm and 60 nm, respectively.

Supplementary (Section 4, page 3):

S4. Insertion loss and associated effective refractive index change

To evaluate the insertion loss per unit length and the corresponding Δn_{eff} , we fabricated a series of waveguide transmission test structures and Mach-Zehnder interferometers (MZIs), as shown in Supplementary Fig. 4. Carbon regions with lengths ranging from 50 to 200 μm , in steps of 50 μm , were deposited under identical conditions across all devices.

The insertion loss was extracted using the waveguide transmission test structures by first normalizing the output of the trimmed arm to the reference arm within the same device, and then comparing the normalized transmission before and after deposition. The waveguide transmission test structures yield a unit-length loss of 0.0078 dB μm^{-1} , while the MZIs fabricated in the same batch give a similar value of 0.0074 dB μm^{-1} for the same deposition lengths, as shown in Supplementary Fig. 5. The close agreement between these measurements confirms the reliability of the extracted loss, whose averaged value of 0.0076 dB μm^{-1} is reported in the main text. The Δn_{eff} was subsequently determined

from the spectral shifts of these MZIs, giving an average value of approximately 0.013.

Supplementary Fig. 4. **a** Optical microscope and SEM views of the waveguide transmission test structure. **b** Optical microscope and SEM views of the MZI used to extract the loss and the effective refractive index change.

Supplementary Fig. 5. **a** Extracted insertion loss for deposited carbon of different lengths on waveguide transmission test structures. **b** Extracted insertion loss obtained from the MZIs. Carbon deposition on both types of structures was carried out under identical deposition conditions.

Main text (last paragraph on page 10):

The A-DC is used as a demonstrator of the trimming concept; the method itself is not restricted to this particular geometry. We also applied the method to a directional coupler and an MZI, both exhibiting the expected response after carbon deposition (Supplementary Fig. 9). With these validations, we next benchmark the trimming performance of our method using the MZI results. The interference-fringe shift provides a direct and quantitative measure of the induced phase change, enabling a fair comparison with representative non-volatile trimming technologies. For clarity, we categorize representative trimming technologies into two groups: invasive and non-invasive, as shown in Table 1. Invasive trimming methods typically induce permanent modifications in the waveguide core material, and the state-of-the-art methods such as ion implantation and laser-based trimming have already achieved very low insertion-loss values of approximately 0.2–0.3 dB per π phase shift^{23,54,55}. In addition, a nearly lossless trimming method has recently been demonstrated using high-temperature annealing of undercut waveguides²⁷. Compared with invasive methods, non-invasive trimming techniques preserve the waveguide core, thereby offering broader platform compatibility. Among these,

programmable PCMs offer large refractive-index modulation through controlled phase transitions and are widely explored for trimming, with recent demonstrations reporting insertion loss of approximately 0.3–0.4 dB per π phase shift^{20,22}. The insertion loss achieved by our method lies within this performance range, supporting its suitability as a practical trimming option.

Table 1 Comparison of state-of-the-art non-volatile trimming techniques.

Invasive type						
Approach	Platform	λ [μm]	IL_{π} [dB]	L_{π} [μm]	$T_{\text{processing}}$ [$^{\circ}\text{C}$]	Ref.
Ion implantation	Si	1.55	$\sim 0.2^{\text{a}}$	$\sim 6^{\text{a}}$	~ 450	55,56
UV laser trimming	SiN	1.95	$\sim 0.28^{\text{a}}$	53^{b}	N.A.	23
fs laser trimming	Si	1.55	$\sim 0.31^{\text{a}}$	N.A.	N.A.	24
Annealing	Si	1.55	~ 0	N.A.	~ 1000	27
Non-invasive type						
Approach	Platform	λ [μm]	IL_{π} [dB]	L_{π} [μm]	$T_{\text{processing}}$ [$^{\circ}\text{C}$]	Ref.
$\text{Ge}_2\text{Sb}_2\text{Te}_5$	Si	2.32	2.6	25	~ 150	21
Sb_2Se_3	SiN	1.55	1.15	15.5	~ 200	57
Sb_2Se_3	Si	1.55	0.38	~ 7	N.A.	22
Sb_2S_3	SiN	1.55	0.89	38.75	~ 300	57
Sb_2S_3	Si	1.55	0.33	24	~ 310	20
FIB carbon deposition	SiN	1.55	0.35	50	RT	^c

^a Estimated based on data reported in the reference.

^b Total length reported for a π phase shift in the parallel-exposure configuration.

^c Data obtained in this work.

N.A.: Not available in the original reference.

Comment 5:

In Supplementary Note S4, it's mentioned that ring resonators were measured with carbon deposition. Through linewidth and resonance wavelength measurements before and after carbon deposition, it should be possible to calculate the change in propagation loss and effective index of the waveguide. It appears possible to extract this information from Fig. S5 and this analysis should be performed.

Our response:

Thank you so much for providing this helpful approach for calculating the parameters mentioned in Comment 4. We have followed this method to obtain the corresponding values and have included them in the Supplementary.

Revision:

Supplementary (on page 5):

To quantify the trimming performance, we evaluate the as-trimmed resonance peak shift of 1.6 nm. With a free-spectral range of 6.5 nm, this corresponds to a phase shift of approximately 0.5π . The relatively thick deposited carbon in this ring resonator leads to an extracted local Δn_{eff} of 0.038. The carbon deposition introduces an additional round-trip

loss of 0.2 dB, corresponding to $0.4 \text{ dB } \pi^{-1}$, which is comparable to the loss-per- π extracted from the MZI in section S7.

Supplementary Fig. 7. **a** A focused ion beam image of a ring resonator after carbon deposition. **b** Transmission peaks of the ring resonator before and after carbon deposition and thermal treatment. The day on which carbon deposition was performed was defined as day one.

Comment 6:

In Fig. 6, trimming of asymmetric directional couplers is performed in a photonic crossbar array circuit. A schematic of the circuit should be provided. The details of this circuit are not clear from Fig. 6(a).

Our response:

We sincerely appreciate this detailed question. We have added the optical microscope image of the photonic crossbar array to the Supplementary.

Revision:

Main text (first paragraph on page 10):

As shown in Fig. 6a, the A-DCs (indicated by a white dashed box) can be added before the output grating coupler to combine the output light of two small 9×2 crossbar arrays, which is equivalent to performing an addition operation (Supplementary Fig. 8 for device layout).

Supplementary (Section S6, page 5):

S6. Layout of the photonic crossbar array with trimmed asymmetric directional couplers

The optical microscope image of the photonic crossbar array presented in the main text is shown in Supplementary Fig. 8. The structure comprises two typical 9×2 photonic crossbar arrays and two A-DCs that combine the output light from the two arrays with minimal loss via mode-division multiplexing. Owing to the flexibility of the FIB carbon deposition, the trimming can be performed directly on the fabricated chip without requiring any predetermined auxiliary structures.

Supplementary Fig. 8. Optical microscope image of the fabricated photonic crossbar array.

Comment 7:

When discussing alternative trimming methods in the introduction, the manuscript states "The trimming method based on micro-heaters usually requires continuous heating to maintain the desired tuning states, resulting in significant power consumption and can cause thermal crosstalk." There are multiple reported methods of thermal trimming where devices are locally annealed using micro-heaters and continuous heating is not required. This statement should be revised and additional references added, for example:

Yanran Xie, Henry C. Frankis, Jonathan D. B. Bradley, and Andrew P. Knights, "Post-fabrication resonance trimming of Si₃N₄ photonic circuits via localized thermal annealing of a sputter-deposited SiO₂ cladding," Opt. Mater. Express 11, 2401-2412 (2021)

Tianyuan Xue, Hannes Wahn, Andrei Stalmashonak, Joyce K. S. Poon, Wesley D. Sacher, "In situ Thermal Trimming of Waveguides in a Standard Active Silicon Photonics Platform," arXiv:2506.13578 [physics.optics] (2025)

Yating Wu, Haozhe Sun, Bo Xiong, Yalong Yv, Jiale Zhang, Zhaojie Zheng, Wei Ma, Tao Chu, "Lossless, Non-Volatile Post-Fabrication Trimming of PICs via On-Chip High-Temperature Annealing of Undercut Waveguides," arXiv:2506.18633 [physics.optics] (2025)

Our response:

Thank you so much for pointing out this important question. We appreciate the opportunity to clarify our intended meaning. Our original description referred to volatile trimming techniques based on microheaters, and we have revised the text to make this point explicit.

Revision:

Main text (second paragraph on page 2):

Various post-fabrication trimming techniques have been proposed to address fabrication imperfections. Volatile approaches are typically based on thermal tuning using microheaters^{14,15}, with the trimmed state sustained while electrical power is applied. In contrast, non-volatile approaches, such as electron beam exposure of polymer claddings^{16–18}, programmable phase-change materials (PCMs) integrated on waveguides^{19–22}, laser-based trimming^{23,24}, high-temperature local annealing^{25–27}, and focused ion beam (FIB) processing^{15,28–31}, permanently modify the optical response without sustained power consumption, but rely on distinct physical mechanisms and are implemented under different processing conditions. Polymer cladding trimming is a simple and cost-effective method, although its long-term stability can be limited. PCM-based methods are broadly compatible with multiple material platforms, but they generally introduce higher insertion loss due to mode mismatch and intrinsic absorption. High-temperature local annealing and ion implantation have demonstrated low loss and high stability. Owing to their thermally driven material modification process, they are most widely used on platforms such as silicon or Si₃N₄.

Reference:

25. Xie, Y., Frankis, H. C., Bradley, J. D. B. & Knights, A. P. Post-fabrication resonance trimming of Si₃N₄ photonic circuits via localized thermal annealing of a sputter-deposited SiO₂ cladding. *Opt. Mater. Express* 11, 2401 (2021).
26. Xue, T., Wahn, H., Stalmashonak, A., Poon, J. K. S. & Sacher, W. D. In situ thermal trimming of waveguides in a standard active silicon photonics platform. *Opt. Express* 33, 43542 (2025).
27. Wu, Y. et al. Lossless Non-Volatile Post-Fabrication Trimming of Silicon Photonics Through On-Chip High-Temperature Annealing of Undercut Waveguides. *Laser Photonics Rev.* e01982 (2025) doi:10.1002/lpor.202501982.

We hope to have addressed all concerns of the referee with the aforementioned revisions to our manuscript and trust that she/he will be in favor of publishing our results in *Nature Communications*.

Manuscript number: NCOMMS-25-65564A

Response to Reviewer 1:

We thank Reviewer 1 for the careful and constructive assessment of our manuscript. The reviewer's rigorous comments helped us clarify the scope and applicability of this work. In response, we have revised the manuscript to more accurately reflect its present level of technological development.

Below is a point-by-point response detailing how each comment has been addressed.

Original comment:

The paper is now significantly improved and can be published after a couple of minor further modifications:

Firstly, the authors express the excess loss due to carbon as $0.0076 \text{ dB } \mu\text{m}^{-1}$. It should be expressed in the more conventional units of dB/cm . i.e. 76 dB/cm . I suspect they want to make the loss appear very low, but I find it misleading, and they can easily follow the value expressed in dB/cm with the insertion loss for a very short section of a few tens of microns, which would still be very low. Comment 1

The authors argue that the method is "well-suited for a wide range of PIC applications demanding precise optical control", and "provides a practical and forward-compatible pathway toward large-scale, energy efficient, and yield-enhancing photonic integration".

However, FIB processing is not really a commercial fabrication process, the use of carbon is problematic as it contaminates fabrication processes, and the method only works for waveguides without cladding, as carbon treatment would preclude these waveguides from going back into a fab for deposition of a cladding. Therefore, I cannot agree that the process is well suited for a wide range of PIC applications, nor that it provides a practical pathway for large scale integration. It does however, suit University fabs, or some research labs that do not have strict contamination policies. Therefore, these phrases should be removed and replaced with more realistic conclusions. Comment 2

We thank the referee for the favorable assessment of our work. Our response to the reviewer's comments are collected in the following.

Comment 1:

Firstly, the authors express the excess loss due to carbon as $0.0076 \text{ dB } \mu\text{m}^{-1}$. It should be expressed in the more conventional units of dB/cm . i.e. 76 dB/cm . I suspect they want to make the loss appear very low, but I find it misleading, and they can easily follow the value expressed in dB/cm with the insertion loss for a very short section of a few tens of microns, which would

still be very low.

Response:

Thank you very much for this important comment. We agree that expressing the excess loss in the conventional unit of dB cm^{-1} improves clarity and avoids potential confusion. We have therefore adopted dB cm^{-1} as the primary metric throughout the manuscript and revised the relevant text accordingly.

In addition, for clarity at the abstract level, we have replaced the unit-length excess loss of 76 dB cm^{-1} with a device-level loss metric expressed as $\text{dB per } \pi$ phase shift for the demonstrated device ($0.35 \text{ dB per } \pi$ phase shift), while retaining dB cm^{-1} as the primary metric in the main text.

Revision:

Main text (abstract on page 1):

Structural characterizations verify the non-invasive nature of the localized carbon deposition, and device measurements show discrete transmission tuning levels of 1.46–16.1 dB. Independent test structures further reveal **an additional loss of 0.35 dB per π phase shift, indicating a low optical penalty at the device level.** Furthermore, the optical response remains stable over two months following a brief initial settling phase.

Main text (first paragraph on page 7):

Since the carbon is not completely lossless, we prepared simple waveguide transmission test structures and Mach-Zehnder interferometers (MZIs) (Supplementary Fig. 4 and Fig. 5) to quantify the insertion loss per unit length and the corresponding change in effective refractive index Δn_{eff} , respectively. The deposited carbon exhibits a unit-length loss of 76 dB cm^{-1} , as shown in Fig. 4a. Under the same deposition conditions, the induced Δn_{eff} is found to be approximately 0.013.

Main text (first paragraph on page 11):

We present a trimming method for PICs based on FIB carbon deposition, which is non-volatile, low-loss, highly stable, room-temperature, and non-invasive. As a representative case, we demonstrate its application in post-fabrication tuning of A-DCs, achieving discrete trimming levels ranging from $\sim 1.5 \text{ dB}$ to 16.1 dB . The unit-length excess loss of the deposited carbon is 76 dB cm^{-1} , and the additional insertion loss introduced during the A-DC trimming is estimated to be $\sim 0.3 \text{ dB}$, indicating the minimal optical penalty associated with this technique.

Comment 2:

The authors argue that the method is "well-suited for a wide range of PIC applications demanding precise optical control", and "provides a practical and forward-compatible pathway

toward large-scale, energy efficient, and yield-enhancing photonic integration".

However, FIB processing is not really a commercial fabrication process, the use of carbon is problematic as it contaminates fabrication processes, and the method only works for waveguides without cladding, as carbon treatment would preclude these waveguides from going back into a fab for deposition of a cladding. Therefore, I cannot agree that the process is well suited for a wide range of PIC applications, nor that it provides a practical pathway for large scale integration. It does however, suit University fabs, or some research labs that do not have strict contamination policies. Therefore, these phrases should be removed and replaced with more realistic conclusions.

Response:

We appreciate this constructive comment regarding the practical scope of the proposed method. We agree that, at its current stage, the approach is primarily suited for research-oriented environments rather than large-scale commercial manufacturing. In response, we have revised the manuscript to remove statements suggesting broad industrial applicability or immediate suitability for large-scale integration, and to provide a more realistic assessment of the method. The corresponding changes have been made in the discussion section.

Revision:

Main text (discussion on page 11):

We present a trimming method for PICs based on FIB carbon deposition, which is non-volatile, low-loss, highly stable, room-temperature, and non-invasive. As a representative case, we demonstrate its application in post-fabrication tuning of A-DCs, achieving discrete trimming levels ranging from ~1.5 dB to 16.1 dB. The unit-length excess loss of the deposited carbon is 76 dB cm⁻¹, and the additional insertion loss introduced during the A-DC trimming is estimated to be ~0.3 dB, indicating the minimal optical penalty associated with this technique. These results validate the method's effectiveness on a fabrication-sensitive component, while its combined advantages render it broadly applicable well beyond A-DCs. The trimmed devices also maintain stable performance over a period exceeding two months, confirming the long-term reliability of the method. Furthermore, the trimming does not rely on localized heating, enabling compact layouts and facilitating dense integration. Because the deposition is performed at room temperature and occurs only on the waveguide surfaces, leaving the waveguide core unaltered, the method inherently preserves device integrity and is therefore, in principle, compatible with both passive and active photonic components, provided the trimmed sections are placed outside the electrically active region. **However, as a FIB-based technique in its present implementation, the proposed approach constitutes a research-stage post-fabrication trimming technique primarily suited for device-level calibration in academic and exploratory fabrication environments.**

Crucially, elemental mapping and EELS plasmon map confirm its non-invasive nature, underpinning broad compatibility across diverse material platforms. Although Ga ions are

detected in the STEM EDS analysis, our EELS measurements show that the deposited carbon exhibits a bandgap of 3.9 eV, well within the accepted range for diamond-like amorphous carbon. This in turn, indicates that Ga content remains below levels that would compromise optical performance, as further supported by the transmission spectra of photonic devices after carbon deposition. Building on these advantages, FIB carbon deposition offers a mask-free solution for post-fabrication tuning in both current silicon-based platforms and emerging material systems. ~~As such, this method provides a practical and forward-compatible pathway toward large-scale, energy-efficient, and yield-enhancing photonic integration across a wide spectrum of applications from high-density interconnects to next-generation optical computing systems.~~

Response to Reviewer 2:

We thank Reviewer 2 for the positive evaluation of our manuscript and for the recommendation.

Original comment:

The authors have addressed all of my comments through additional characterization work and clarifications. I recommend that the manuscript be accepted for publication.

Manuscript number: NCOMMS-25-65564B

Response to Reviewer 1:

We thank Reviewer 1 for the careful assessment of our manuscript. The reviewer's comments have contributed to improving the precision of the wording and clarifying the scope of this work. The continued discussion with Reviewer 1 has been helpful in guiding the future development of the technique presented in this work. In response, we have revised the manuscript to remove or modify statements that could be interpreted as overstating the applicability of the technique.

Below, we provide a point-by-point response describing how each comment has been addressed.

Original comment:

This is the second review following a previous review and corrections by the authors. The paper is indeed much better, and the authors have addressed some of my concerns. However, I believe they have not addressed all of my concerns. They are still overclaiming the impact of the technology. They have made much of the additional 2 paragraphs showing that the technique is only suitable for research environments, which are welcome. However, in the abstract a sentence remains that says "These results establish FIB carbon deposition as a robust and broadly applicable PIC trimming technique...."

These results do not do that. They show that the technique has potential, but they have not demonstrated that it is either widely applicable nor robust. If it were widely applicable it would be able to be used widely in fabrications facilities, but I doubt any commercial facility will allow wafers with Carbon on the surface to be processed. That means that claddings cannot be deposited, and almost no commercial PICS would be used without a cladding. The authors mention in passing some results with a cladding, but these are not discussed in any meaningful way in the paper. Secondly, I suspect the word "robust" is used because they have monitored the devices over 2 months. This does not mean that the samples are stable over years, as is effectively claimed. They have no idea what would happen under accelerated aging. That does not mean that the work is without promise and value, but it does mean that they should not claim things that have not been demonstrated. This is particularly pertinent because in the second paragraph of the introduction, they criticise PCM based methods by saying ".....although its long-term stability can be limited." Comment 1

In the next paragraph they claim that Fib technology is widely available in advanced manufacturing laboratories. Which implies it is an established manufacturing technique for PICS. This is not true, and is not compatible with my previous comments about the commercial applicability of the technology, which they say they agree with in the attached letter. Comment 2

In the second paragraph of page 3 they say "we demonstrate FIB carbon deposition as a non-

invasive technique with great stability". As discussed above, they have not demonstrated this. They have merely demonstrated stability in a period between 2 weeks and 2 months. ^{Comment 3}

Then there is a whole discussion about different technologies that they describe as invasive or non invasive, classifying their technology as non invasive, which suggests it is somehow better than what they call invasive technologies/ However, these definitions are not appropriate or correct. The cladding of a waveguide is an integral part of the waveguide structure and their results are based on waveguides for which a cladding cannot be used, except in contaminated fabrication facilities, and even then they have not demonstrated a cladding on top of their carbon deposition, and therefore have not assessed its impact on the integrity of any cladding. Thus I cannot accept that their technique is non invasive, unless they actually demonstrate that. Again they are over-claiming conclusions. These claims are repeated in the discussion section on page 11 ("broadly applicable", "long term stability", "preserves device integrity".

Comment 4

Therefore, I'm afraid the paper is still not suitable for publication until the unjustified claims are removed.

We acknowledge the reviewer's concerns and have addressed them in detail below.

Comment 1:

This is the second review following a previous review and corrections by the authors. The paper is indeed much better, and the authors have addressed some of my concerns. However, I believe they have not addressed all of my concerns. They are still overclaiming the impact of the technology. They have made much of the additional 2 paragraphs showing that the technique is only suitable for research environments, which are welcome, However, in the abstract a sentence remains that says "These results establish FIB carbon deposition as a robust and broadly applicable PIC trimming technique....".

These results do not do that. They show that the technique has potential, but they have not demonstrated that it is either widely applicable nor robust. If it were widely applicable it would be able to be used widely in fabrications facilities, but I doubt any commercial facility will allow wafers with Carbon on the surface to be processed. That means that claddings cannot be deposited, and almost no commercial PICS would be used without a cladding. The authors mention in passing some results with a cladding, but these are not discussed in any meaningful way in the paper. Secondly, I suspect the word "robust" is used because they have monitored the devices over 2 months. This does not mean that the samples are stable over years, as is effectively claimed. They have no idea what would happen under accelerated aging. That does not mean that the work is without promise and value, but it does mean that they should not claim things that have not been demonstrated. This is particularly pertinent because in the second paragraph of the introduction, they criticise PCM based methods by saying ".....although its long-term stability can be limited."

Response:

The wording in the abstract has been revised following the reviewer's comments to avoid any overstatement of the scope. Specifically, the phrase "robust and broadly applicable" has been removed, and the abstract now reflects only the demonstrated results and potentials.

We have also revised the sentence to clarify that the statement on limited long-term stability refers to polymer-based trimming methods rather than PCM-based approaches.

Revision:

Main text (abstract on page 1):

Photonic integrated circuits (PICs), widely used in optical communications and computing, require precise post-fabrication trimming due to their high sensitivity to fabrication imperfections. Focused ion beam (FIB) carbon deposition offers a localized trimming approach with high spatial precision. Here, we demonstrate this technique for the first time to enable non-volatile post-fabrication trimming of PICs. To validate this approach, we use asymmetric directional couplers as representative fabrication-sensitive components. Structural characterizations confirm localized surface deposition without observable modification of the underlying waveguide core, and device measurements show discrete transmission tuning levels of 1.46–16.1 dB. Independent test structures further reveal an additional loss of 0.35 dB per π phase shift, indicating a low optical penalty at the device level. Furthermore, the optical response remains stable over two months following a brief initial settling phase. ~~These results establish FIB carbon deposition as a robust and broadly applicable PIC trimming technique, empowering optical communications, computing, and technologies beyond to meet rising data demands.~~ These results highlight the potential of FIB carbon deposition in device-level trimming and provide a foundation for exploring future trimming strategies toward parallel implementations.

Main text (second paragraph on page 2):

Polymer cladding trimming is a simple and cost-effective method, although its long-term stability can be limited ~~depending on environmental conditions~~. PCM-based methods are broadly compatible with multiple material platforms, but they generally introduce higher insertion loss due to mode mismatch and intrinsic absorption.

Comment 2:

In the next paragraph they claim that Fib technology is widely available in advanced manufacturing laboratories. Which implies it is an established manufacturing technique for PICS. This is not true, and is not compatible with my previous comments about the commercial applicability of the technology, which they say they agree with in the attached letter.

Response:

We thank the Reviewer for pointing out that the phrase "advanced manufacturing laboratories"

could be misinterpreted. To avoid any ambiguity, we have removed this phrase from the manuscript. The revised text now refers only to the availability of FIB in research facilities.

Revision:

Main text (last paragraph on page 2):

Given that FIB technology is well-established and widely available in research facilities ~~and advanced manufacturing laboratories~~, this presents a valuable opportunity for precise post-fabrication trimming in PICs.

Comment 3:

In the second paragraph of page 3 they say "we demonstrate FIB carbon deposition as a non-invasive technique with great stability". As discussed above, they have not demonstrated this. They have merely demonstrated stability in a period between 2 weeks and 2 months.

Response:

We have revised the manuscript in response to the Reviewer's comment regarding the stability claim to more precisely describe the results as "stability over a two-month observation period", replacing the previous broader wording. The revised text is highlighted in the manuscript.

Revision:

Main text (second paragraph on page 3):

In this study, we demonstrate FIB carbon deposition as a localized trimming technique for PICs. To validate its effectiveness, we employ A-DCs as a sensitive test structure, where trimming allows precise control of device responses. We also quantify the trimming-induced insertion loss, thermal behavior, accessible tuning range, and stability over a two-month observation period, demonstrating the potential of this approach for high-resolution trimming in representative PIC devices.

Comment 4:

Then there is a whole discussion about different technologies that they describe as invasive or non invasive, classifying their technology as non invasive, which suggests it is somehow better than what they call invasive technologies/ However, these definitions are not appropriate or correct. The cladding of a waveguide is an integral part of the waveguide structure and their results are based on waveguides for which a cladding cannot be used, except in contaminated fabrication facilities, and even then they have not demonstrated a cladding on top of their carbon deposition, and therefore have not assessed its impact on the integrity of any cladding. Thus I cannot accept that their technique is non invasive, unless they actually demonstrate that. Again they are over-claiming conclusions. These claims are repeated in the discussion section on page 11 ("broadly applicable", "long term stability", "preserves device integrity".

Response:

We thank the Reviewer for the careful comments regarding the use of the terms "invasive" and "non-invasive." To avoid possible misunderstanding, we have removed this terminology from the manuscript and revised the related discussion accordingly. In addition, statements regarding broad applicability, long-term stability, and device integrity have been removed so that the text reflects only the demonstrated experimental results.

Revision:

Main text (last paragraph on page 10):

The A-DC is used as a demonstrator of the trimming concept; the method itself is not restricted to this particular geometry. We also applied the method to a directional coupler and an MZI, both exhibiting the expected response after carbon deposition (Supplementary Fig. 9). With these validations, we next benchmark the trimming performance of our method using the MZI results. The interference-fringe shift provides a direct and quantitative measure of the induced phase change, enabling a fair comparison with representative non-volatile trimming technologies. Table 1 summarizes selected approaches reported in the literature. Methods based on permanent modifications in the waveguide core material, such as ion implantation and laser-based trimming, have already achieved very low insertion-loss values of approximately 0.2–0.3 dB per π phase shift^{23,54,55}. In addition, a nearly lossless trimming method has recently been demonstrated using high-temperature annealing of undercut waveguides²⁷. PCM-based trimming approaches provide large refractive-index modulation through controlled phase transitions and are widely explored for trimming, with recent demonstrations reporting insertion loss of approximately 0.3–0.4 dB per π phase shift^{20,22}. The insertion loss achieved by our method lies within this performance range, supporting its suitability as a device-level trimming option.

Table 1 Comparison of state-of-the-art non-volatile trimming techniques.

Approach	Platform	λ [μm]	IL_{π} [dB]	L_{π} [μm]	$T_{\text{processing}}$ [$^{\circ}\text{C}$]	Ref.
Ion implantation	Si	1.55	$\sim 0.2^{\text{a}}$	$\sim 6^{\text{a}}$	~ 450	55,56
UV laser trimming	SiN	1.95	$\sim 0.28^{\text{a}}$	53 ^b	N.A.	23
fs laser trimming	Si	1.55	$\sim 0.31^{\text{a}}$	N.A.	N.A.	24
Annealing	Si	1.55	~ 0	N.A.	~ 1000	27
$\text{Ge}_2\text{Sb}_2\text{Te}_5$	Si	2.32	2.6	25	~ 150	21
Sb_2Se_3	SiN	1.55	1.15	15.5	~ 200	57
Sb_2Se_3	Si	1.55	0.38	~ 7	N.A.	22
Sb_2S_3	SiN	1.55	0.89	38.75	~ 300	57
Sb_2S_3	Si	1.55	0.33	24	~ 310	20
FIB carbon deposition	SiN	1.55	0.35	50	RT	^c

^a Estimated based on data reported in the reference.

^b Total length reported for a π phase shift in the parallel-exposure configuration.

^c Data obtained in this work.

N.A.: Not available in the original reference.

Main text (Discussion on page 11):

We present a trimming method for PICs based on FIB carbon deposition, which is non-volatile, low-loss, deposited at room-temperature, and stable over a two-month observation period. As a representative case, we demonstrate its application in post-fabrication tuning of A-DCs, achieving discrete trimming levels ranging from ~ 1.5 dB to 16.1 dB. The unit-length excess loss of the deposited carbon is 76 dB cm^{-1} , and the additional insertion loss introduced during the A-DC trimming is estimated to be ~ 0.3 dB, indicating the minimal optical penalty associated with this technique. These results demonstrate the method's feasibility on a fabrication-sensitive component, ~~while its combined advantages render it broadly applicable well beyond A-DCs. The trimmed devices also maintain stable performance over a period exceeding two months, confirming the long-term reliability of the method.~~ Furthermore, the trimming does not rely on localized heating, enabling compact layouts and facilitating dense integration. Because the deposition is performed at room temperature and confined to the waveguide surfaces, leaving the waveguide core unaltered, the method can in principle be applied inherently preserves device integrity and is therefore, in principle, compactible with to both passive and active photonic components, provided the trimmed sections are placed outside the electrically active regions. However, as a FIB-based technique in its present implementation, the proposed approach constitutes a research-stage post-fabrication trimming technique primarily suited for device-level calibration in academic and exploratory fabrication environments.

Crucially, elemental mapping and EELS plasmon map confirm localized surface deposition in the demonstrated structures. Although Ga ions are detected in the STEM EDS analysis, our EELS measurements show that the deposited carbon exhibits a bandgap of 3.9 eV, well within the accepted range for diamond-like amorphous carbon. This in turn, indicates that Ga content remains below levels that would compromise optical performance, as further supported by the transmission spectra of photonic devices after carbon deposition. Building on these advantages, FIB carbon deposition provides a mask-free trimming approach for the demonstrated SiN-based platform, and has potentials to be extended to other material systems subject to further verification. Together with the device-level optical measurements, these microstructural insights provide a consistent physical picture of the deposition process and provide the foundation for further studies in relation to more complex photonic devices.